# ADGym: Design Choices for Deep Anomaly Detection

**Minqi Jiang**[1,*], **Chaochuan Hou**[1,*], **Ao Zheng**[1,*] **Songqiao Han**[1,†],
**Hailiang Huang**[1,2,†], **Qingsong Wen**[3], **Xiyang Hu**[4,†], **Yue Zhao**[4,†]

[1]AI Lab, Shanghai University of Finance and Economics
[2]MoE Key Laboratory of Interdisciplinary Research of Computation and Economics
[3]DAMO Academy, Alibaba Group [4]Carnegie Mellon University
jiangmq95@163.com, houchaochuan@foxmail.com, zheng-ao@outlook.com,
{han.songqiao,hlhuang}@shufe.edu.cn, qingsongedu@gmail.com,
{xiyanghu,zhaoy}@cmu.edu

## Abstract

Deep learning (DL) techniques have recently found success in anomaly detection (AD) across various fields such as finance, medical services, and cloud computing. However, most of the current research tends to view deep AD algorithms as a whole, without dissecting the contributions of individual design choices like loss functions and network architectures. This view tends to diminish the value of preliminary steps like data preprocessing, as more attention is given to newly designed loss functions, network architectures, and learning paradigms. In this paper, we aim to bridge this gap by asking two key questions: *(i)* Which design choices in deep AD methods are crucial for detecting anomalies? *(ii)* How can we automatically select the optimal design choices for a given AD dataset, instead of relying on generic, pre-existing solutions? To address these questions, we introduce ADGym, a platform specifically crafted for comprehensive evaluation and automatic selection of AD design elements in deep methods. Our extensive experiments reveal that relying solely on existing leading methods is not sufficient. In contrast, models developed using ADGym significantly surpass current state-of-the-art techniques.

## 1 Introduction

Anomaly detection (AD) aims to identify data objects that significantly deviate from the majority of samples, with numerous successful applications in intrusion detection [36, 44], fault detection [22, 85], medical diagnosis [13, 38], fraud detection [2, 8, 9], social media analysis [81, 86], etc. Recently, deep neural networks have become the primary techniques in AD due to their powerful representation learning capacity [57]. Among all, weakly-supervised AD (WSAD) methods [56, 58, 59, 69, 83, 90], which leverage imperfect ground truth anomaly labels (e.g., those that are incomplete, inaccurate, or inexact) in deep neural networks for AD, have gained attention in this new frontier [29]. A comprehensive study by [26] showcases WSAD's superiority over unsupervised AD techniques, especially for real-world conditions where the ground truth labels are never complete and accurate. Thus, our work introduces ADGym, a platform for understanding and designing deep WSAD methods[2], with the potential to be extended to unsupervised and supervised deep AD methods.

**Goal I: Understanding Design Choices of Deep AD.** Many WSAD methods attribute their improvements to novel network architectures or loss functions, based on the authors' understanding of anomalies. However, these choices represent only a fraction of the design considerations, as there are many other factors to consider, such as data preprocessing and model training techniques. In Table

---

[*]Contribute equally. [†]Corresponding authors.
[2]For brevity, AD methods in this work refer to deep WSAD methods.

37th Conference on Neural Information Processing Systems (NeurIPS 2023) Track on Datasets and Benchmarks.

Table 1: ADGym supports a comprehensive list of design choices for deep AD methods.

| Pipeline | Design Dimensions | Design Choices |
|---|---|---|
| Data Handling | **Data Augmentation** | [Oversampling, SMOTE, Mixup, GAN] |
| | Data Preprocessing | [MinMax, Normalization] |
| Network Construction | **Network Architecture** | [MLP, AutoEncoder, ResNet, FTTransformer] |
| | Hidden Layers | [[20], [100, 20], [100, 50, 20]] |
| | **Activation** | [Tanh, ReLU, LeakyReLU] |
| | Dropout | [0.0, 0.1, 0.3] |
| | Initialization | [default, Xavier (normal), Kaiming (normal)] |
| Network Training | **Loss Function** | [BCE, Focal, Minus, Inverse, Hinge, Deviation, Ordinal] |
| | **Optimizer** | [SGD, Adam, RMSprop] |
| | Epochs | [20, 50, 100] |
| | Batch Size | [16, 64, 256] |
| | **Learning Rate** | [1e-2, 1e-3] |
| | Weight Decay | [1e-2, 1e-4] |

*Note*: *Bolded design dimensions are those of greater impact on the AD task (discussed in detail in §4).*

1, we list specific design choices for deep AD models and group them as design dimensions. Our experiments reveal that some of these dimensions (in bold) significantly impact the performance.

Many questions remain unanswered in current AD studies, such as the interaction between components within an AD method, the relative importance of each component in performance, and the potential of new data-centric techniques to improve AD model performance. Thus, in the first part of the study (§3.2), we have evaluated various design combinations on large benchmark datasets. Interestingly, the optimal model, comprising different design choices by combination, varies by datasets and notably outperforms existing state-of-the-art (SOTA) AD models. This raises a key question: how can we automatically design AD models for datasets other than using existing models?

**Goal II: Constructing AD Algorithms Automatically via ADGym.** Indeed, our prior research [88, 53, 79] also shows there is no one-size-fits-all AD model for every dataset—we must select AD models based on the underlying dataset. Our prior work focuses on model selection from a pre-defined list, mainly for non-neural-network AD methods with limited design choices [88, 89]. In today's deep learning era, a pre-defined list is not sufficient, given the large number of design choices in Table 1 and §3.2). There could be infinite deep AD models with different design combinations.

In the second part of this study, we develop a *design-choice-level* selection approach that utilizes *meta-learning*, called ADGym (see §3.3). In a nutshell, our approach leverages the supervision signals from historical datasets to guide selecting the best design choices for constructing AD models tailored to a new dataset. We aim to enhance the existing AD model selection process, ensuring the best combination of design choices for a given application or dataset. What sets ADGym apart from prior works is our focus on granular design choice selection tailored for deep AD, which has a much larger space than a pre-defined model list for existing methods only. See Fig. 1 for a comparison.

**What Do We Learn from the Experiments (§4)?** ADGym helps us better understand and design deep AD methods, with key observations from our experiments: (1) No single design choice consistently outperforms others across all datasets, justifying the need for an automated approach to collectively select the most effective design choices. (2) Employing a *meta*-predictor to automate design choice selection yields notable improvements over static SOTA methods. (3) We can make meta-predictors better by using ensemble techniques or increasing the range of design choices.

**To sum up, our work makes the following technical contributions**:

1. **Understanding AD Design Choices via Benchmarking.** We present the first benchmark that breaks down and compares diverse deep AD design choices across 29 real-world datasets and 4 groups of synthetic datasets[1], which leads to a few interesting observations.
2. **Design Choice Automation.** We introduce ADGym, the first automated framework for selecting design choices for weakly-supervised AD, which significantly outperforms SOTA AD methods.
3. **Accessibility and Extensibility.** We have made ADGym publicly available[2] so that practitioners can build better-than-SOTA AD methods. It is also easy to include new design choices.

---

[1]348 datasets generated with different seeds to simulate 4 types of anomalies. See Appx. D.2 for details.
[2]The code is available at `https://github.com/Minqi824/ADGym`

# 2 Related Work

## 2.1 Weakly-supervised Anomaly Detection (WSAD)

Due to the cost and difficulties in data annotation, previous studies [42, 43, 47, 68, 91] mainly focus on developing unsupervised AD methods with different assumptions of data distribution [1], while they are shown to have no statistically significant difference from each other in a recent benchmark [26]. In practice, however, there could exist at least a handful of labeled instances that are either identified by domain experts or the bad cases occurred during the deployment of AD methods. Therefore, recent studies [56, 58, 59, 69, 90] propose weakly-supervised AD methods to effectively leverage the valuable knowledge in labeled data and thus facilitate anomaly identification. Nevertheless, existing WSAD methods are often used as *it is*, without detailed evaluation of each design choice like network architectures and loss functions. A more granular analysis of specific components of WSAD methods could be helpful for a deeper understanding of the AD methods.

## 2.2 Benchmarks for Anomaly Detection

Anomaly detection benchmarks mainly perform large comparisons and evaluations on the detection performance of different AD methods under unified and controlled conditions. While previous studies [12, 17, 23, 71, 72] focus on benchmarking classical machine learning AD methods, there has been an increasing trend towards benchmarking deep learning AD methods as well. [67] reviews both classic and deep AD models and points to the connections between classic shallow algorithms and deep AD models. Our previous work [26] performs the largest-scale AD benchmark so far, evaluating 30 shallow and deep AD algorithms across 57 benchmark datasets under different levels of supervision. Besides, some benchmark works focus on AD tasks with different data modalities, including time-series [46, 60, 35], graph [49, 48], CV [78] and NLP [63]. All of these benchmarks focus on discovering which AD algorithms (as a whole) are more effective. Differently, we focus on understanding the effectiveness of design choices in deep AD methods, complementing these works.

## 2.3 Automatic Model Selection for AD

### 2.3.1 Unsupervised Anomaly Model Selection

Developing automatic model selection schemes for unsupervised AD algorithms faces a major challenge in lacking evaluation criteria. A recent survey [53] provides a comprehensive survey of using internal model evaluation strategies that solely rely on input features and outlier scores for model selection. Their extensive experiment shows that *none of the existing and adapted evaluation strategies would be practically useful* for unsupervised AD model selection.

Another existing work stream leverages meta-learning, based on the similarity between the new task and historical datasets where model performance is available. Some notable work [88, 89, 87] assume that model performance can be generalized across similar datasets, thus selecting the model for a new task based on similar historical tasks. Specifically, [89, 87] introduce the idea of building a *meta-predictor* (which is a regressor) to predict the model performance on a new dataset by training it on historical model performances. In ADGym, we take the idea of meta-learning to train a specialized meta-predictor for deep AD methods, where existing works [88, 89] only focus on non-neural-network-based AD models with a relatively small model pool, where we extend to more complex deep AD design spaces. See §3.3 for details.

### 2.3.2 Supervised Anomaly Model Selection, HP Optimization, and Neural Arch. Search

Supervised AD model selection and HP optimization (HPO) aim to train a set of models, evaluate their performance using ground truth labels, and ultimately select the best model. However, as previously mentioned, ground truth labels in AD are scarce, which prevents supervised selection from being widely explored. PyODDS [41] optimizes an AD pipeline, while TODS [34] creates a selection system for time series data. Both, however, focus primarily on shallow models, excluding most deep models. Recently, AutoOD [40] conducts neural architecture search (NAS) for deep AD using labels. However, it only focuses on AutoEncoder models over image data. AutoPatch[31] and [73] are another two NAS works for AD, but they still do not consider networks beyond CNNs.

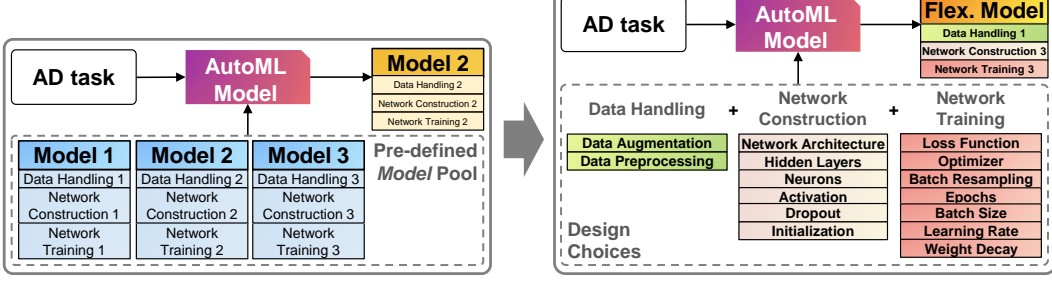

Figure 1: The framework of ADGym. Existing AD model selection focuses on selecting the best model from a small, fixed pool. Our proposed method is more flexible to build a good model by choosing different parts, including data handling, network construction, and network training.

Here, we highlight the difference between ADGym and the above works. First, ADGym aims to select from a large, comprehensive design pool, as opposed to existing model selection, which is often restricted to a *small* pre-defined model pool; meanwhile, ADGym also brings more granularity than general HPO and NAS, which only focus on a small set of HPs (given neural architectures can also be considered as HPs). Second, ADGym supports more scenarios other than focusing on a single family of AD methods, e.g., autoencoders, or just network architectures. Third, different from the SOTA method AutoOD, ADGym is a zero-shot algorithm that does not require any model building for a new dataset, thereby significantly reducing the online time; see §3.3 for details.

## 3 ADGym: *Benchmarking* and *Automating* Design Choices in Deep AD

### 3.1 Problem Definition

In this paper, we focus on the common weakly supervised AD scenario, where given a training dataset $\mathcal{D} = \left\{ \boldsymbol{x}_1^u, \ldots, \boldsymbol{x}_k^u, \left( \boldsymbol{x}_{k+1}^a, y_{k+1}^a \right), \ldots, \left( \boldsymbol{x}_{k+m}^a, y_{k+m}^a \right) \right\}$ contains both unlabeled samples $\mathcal{D}_u = \{ \boldsymbol{x}_i^u \}_{i=1}^k$ and a handful of labeled anomalies $\mathcal{D}_a = \left\{ \left( \boldsymbol{x}_j^a, y_j^a \right) \right\}_{j=1}^m$, where $\boldsymbol{x} \in \mathbb{R}^d$ represents the input feature and $y_j^a$ is the label of identified anomalies. Usually, we have $m \ll k$, since only limited prior knowledge of anomalies is available. Such data assumption is more practical for AD problems, and has been studied in recent deep AD methods [56, 58, 59, 69, 90]. The goal of a weakly-supervised AD model $M$ is to assign higher anomaly scores to anomalies.

### 3.2 Goal I: Understanding Design Choices of Deep AD

The first primary goal of this study is to investigate the large design space of deep AD solutions, which should cover as many design choices as possible, e.g., different data augmentation methods, network architectures, etc. Following the taxonomy of the previous study [80] and practical experiences in industrial applications [2, 22, 78], we decouple the standard process of deep AD methods, starting from the input data and ending with model training. The **Pipeline** includes: *data handling → network construction → network training*, as shown in Table 1. For each step of the pipeline, we further categorize it as different **Design Dimensions**, where diverse **Design Choices** can be specified to instantiate an AD method. More details are shown in Appx. C.1

With the comprehensive design space above, we not only evaluate the possible designs of weakly-supervised AD methods, but also investigate several interesting design dimensions often overlooked in previous AD research. For example, previous AD studies often perform model training on the raw input data (instead of augmented data for mitigating class imbalance problems), and simple network architectures like multi-layer perceptron (MLP) (instead of more recent network architectures like ResNet [27] and Transformer [75]), where the other cutting-edge techniques have already been widely used in other domains like NLP and CV. To sum up, we pair the combination of comprehensive design choices in Table 1 with benchmark AD datasets to unlock new insights. See results in §4.2.

### 3.3 Goal II: Constructing AD Algorithms Automatically via ADGym

**From Model Selection to Pipeline Selection**. With the large AD design space illustrated in the last section, we investigate how to construct effective AD methods given the downstream applications

automatically. Given a pre-defined model set $\mathcal{M} = \{M_1, ..., M_m\}$ that includes $m$ combinations of applicable design choices, model selection picks a model $M \in \mathcal{M}$ to train on a dataset $\mathcal{D}_{test}$ and output the anomaly score $O := M(\mathbf{X}_{\text{test}})$ to achieve the highest detection performance. In our case, we construct a *pipeline* $P$ from our design space to generate each model $M$. As shown in Fig. 1, our design (right) brings more flexibility by choosing from details than existing model selection works that only choose from a fixed, small set of AD models.

**Meta-learning for Pipeline Selection**. Meta learning has been recently used in unsupervised AD model selection [88, 89], where the core idea is to transfer knowledge from model performance information on historical datasets to the selection on a new dataset. Intuitively, if two datasets are similar, their best models should also resemble. Under the meta-learning framework, we assume there are $n$ historical AD datasets (i.e., detection tasks) $\mathcal{D}_{\text{train}} = \{\mathcal{D}_1, \ldots, \mathcal{D}_n\}$; each meta-train dataset is accompanied with ground truth labels for performance evaluation. To leverage prior knowledge for pipeline selection on a new dataset, we first conduct experiments on $n$ historical datasets to evaluate and collect the performance of $m$ possible *designed pipelines* (as the combination of all applicable design choices), by two widely used metrics: AUCROC (Area Under Receiver Operating Characteristic Curve) and AUCPR (Area Under Precision-Recall Curve).

For each metric, we acquire the corresponding performance matrix $\boldsymbol{P} \in \mathbb{R}^{n \times m}$, where $\boldsymbol{P}_{i,j}$ corresponds to the $j$-th constructed AD model's performance on the $i$-th historical dataset. Meanwhile, historical datasets vary in task difficulty, resulting in variations in the numerical range of the constructed AD models' performance. Therefore, we convert the value of the performance into their relative/normalized ranking, where $\boldsymbol{P}_{i,j} = rank(P_{i,j})/m \in [0, 1]$. Smaller ranking values indicate better performance on the corresponding dataset.

To predict the performance of a given pipeline on a new dataset, we propose to train a *meta-predictor* as a *regression* problem: the input of meta-predictor is $\{\mathbf{E}_i^{meta}, \mathbf{E}_j^{comp}\}$, corresponding to the meta-feature [88] (i.e., the unified representations of a dataset) of $i$-th dataset and the embedding of $j$-th AD component (i.e., the representation of a pipeline/components). We defer the specific details into Appx. C.2. Given the meta-predictor $f(\cdot)$, we train it to map dataset and pipeline characteristics to their corresponding performance ranking across all historical datasets, as shown in Eq. (1). Refer to our open-sourced code for details of embeddings and the choice of regression models.

$$ f \; : \; \underbrace{\mathbf{E}_i^{meta}}_{\text{meta features}} \; , \; \underbrace{\mathbf{E}_j^{comp}}_{\text{component embed.}} \; \mapsto \boldsymbol{P}_{i,j} \; , \; i \in \{1, \ldots, n\}, \; j \in \{1, \ldots, m\} \qquad (1) $$

For a newcoming dataset (i.e., test dataset $\mathbf{X}_{\text{test}}$), we acquire the predicted relative ranking of different AD components using the trained $f(\cdot)$, and select top-1 ($k$) to construct AD model(s). Note this procedure is zero-shot without needing any neural network training on $\mathbf{X}_{\text{test}}$ but only extracting meta-features and pipeline embeddings. We show the effectiveness of the meta-predictor in §4.3.

## 4 Experiments

### 4.1 Experiment Settings

**AD Datasets and Baselines**. In ADGym, we gather datasets from the previous AD studies and benchmarks [12, 19, 26, 57, 64] for evaluating existing AD models and different AD designs. Datasets with sample sizes smaller than 1000, as well as those with problematic model results, are removed, resulting in a total of 29 remaining datasets. These datasets cover many diverse application domains, such as healthcare, audio and language processing, image processing, finance, etc. [1] Furthermore, based on our previous work[26], we categorize anomalies into four types, local, global, cluster and dependency anomalies, and generate synthetic datasets for each individual anomaly type. For each dataset, $70\%$ data is split as the training set and the remaining $30\%$ is used as the test set. We use stratified sampling to keep the anomaly ratio consistent. For each experiment, we repeat 3 times and report the average. Further details of datasets and baselines are presented in Appx. A and B.

**Meta-predictor in ADGym**. We introduce two types of meta-predictors in ADGym, namely DL-based and ML-based. The DL-based meta-predictor is instantiated as a two-layer MLP and trained for 100 epochs with early stopping. The training process utilizes the Adam optimizer [32] with a

---

[1]We analyze the results across domains and find no single set of best design choices. See Appx.D.3

learning rate of 0.001 and batch size of 512. For the ML-based meta-predictor, We instantiate the meta-predictor with XGBoost and CatBoost and their default hyperparameter settings. Considering the expensive computational cost of training meta-predictors on the entire design space, we randomly sample 1,000 design choices for each experiment illustrated below. See details in Appx. C.2.

**Evaluation Metrics**. We perform evaluations with two widely used metrics: AUCROC (Area Under Receiver Operating Characteristic Curve), AUCPR (Area Under Precision-Recall Curve) value, and the relative rankings corresponding to these two metrics[1].

## 4.2 Large Evaluation on AD Design Choices

In this work, we perform large evaluations on the decoupled pipelines according to the standard procedure of AD methods. Such analysis is often overlooked in previous AD studies, and we investigate each design dimension of decoupled pipelines by fixing its corresponding design choice (e.g., Focal loss), and randomly sampling other dimensional design choices to construct AD models, e.g., $\{DateAugmentation : Mixup, NetworkArchitecture : FTT, LossFunction : Focal, ...\}$, ..., $\{DateAugmentation : GAN, NetworkArchitecture : MLP, LossFunction : Focal, ...\}$. In other words, we investigate each design dimension by controlling other variables. Different design choices are compared w.r.t. $n_a = 5, 10, 20$ and demonstrated with box plots, where the number of comparisons in each box is ensured to be the same. In the following subsections, we analyze the benchmark results on each of the design dimensions, namely data handling, network construction, and network training.

### 4.2.1 Data Handling

For the design dimension of data augmentation methods (e.g., SMOTE[2] and Mixup methods) shown in Figure 2, we find that *almost no method* has brought significant performance improvements. This could be explained by the fact that, unlike time series [76], NLP [70], or CV tasks [20], data augmentation is rarely incorporated into the design of existing AD methods tailored for tabular data [57]. Besides, our results indicate that GAN-based augmentation method is even worse than simpler methods like oversampling, probably due to the difficulty of modeling and generating realistic anomalies for the tabular data. The same trend is observed in the synthetic datasets with individual anomalies. In the vast majority of cases, the results from data augmentation are inferior to those from the original data. (See Appx.D.2.) For data preprocessing methods, We do not observe a significant difference between the minmax scaling and normalizing methods.

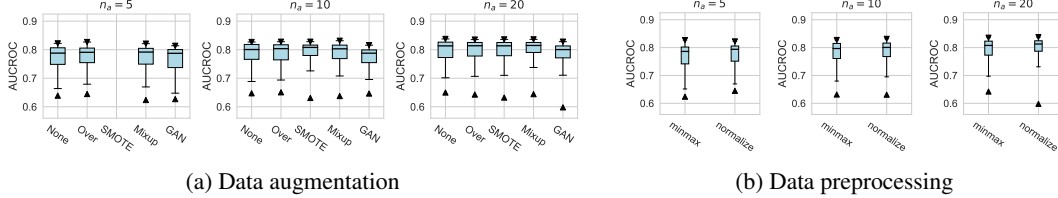

|                | (a) Data augmentation | (b) Data preprocessing |
| :-: | :-: | :-: |

Figure 2: AUCROC performance of data handling designs. There is no significant difference between different design choices in both data augmentation and preprocessing dimensions.

### 4.2.2 Network Construction

We investigate various design dimensions of network construction, as is shown in Figure 3. For network architectures, MLP is *still* a competitive baseline in AD tasks, and even outperforms other more complex architecture designs like ResNet and FTTransformer w.r.t. $n_a = 5$, i.e., only a handful of labeled anomalies are available in the training stage. Moreover, we suggest that the ResNet model could also serve as both an effective and stable network architecture, especially when more labeled anomalies can be acquired (e.g., $n_a = 10$ and $n_a = 20$). However, we do not find significant advantages of the FTTransformer, where its AUCROC performance is generally worse than other network architectures. This may be due to the reason that compared to the supervised tasks for tabular

---

[1] Due to page limit, we report the AUCPR and the relative ranking results in the Appx. D.1.2. We find similar conclusions can be drawn between AUCROC and AUCPR metrics.

[2] For $n_a = 5$, SMOTE raises an error since the number of labeled anomalies is smaller than the neighbors.

data, very limited labeled data (corresponding to the noises in the unlabeled data) can be leveraged to guide the training process of such a complicated model. Furthermore, compared to our earlier work on ADBench[26] where the FTTransformer shows competitive performance, it can be reasonably inferred that the FTTransformer has limited robustness to hyperparameter tuning, especially after we dissected its parameters. We also identify the advantage of the AutoEncoder architecture on synthetic datasets with individual anomaly types, particularly in the local and global anomaly datasets (see Appx. D). A plausible explanation is that the normal data in these synthesized datasets follows a specific distribution that can be more effectively reconstructed.

For the activation function, Tanh and LeakyReLU appear to be more effective than the ReLU function in tabular AD tasks, where this conclusion holds true for different $n_a$. Besides, we do not observe significant differences in the design dimensions of hidden layers, dropout, and network initialization.

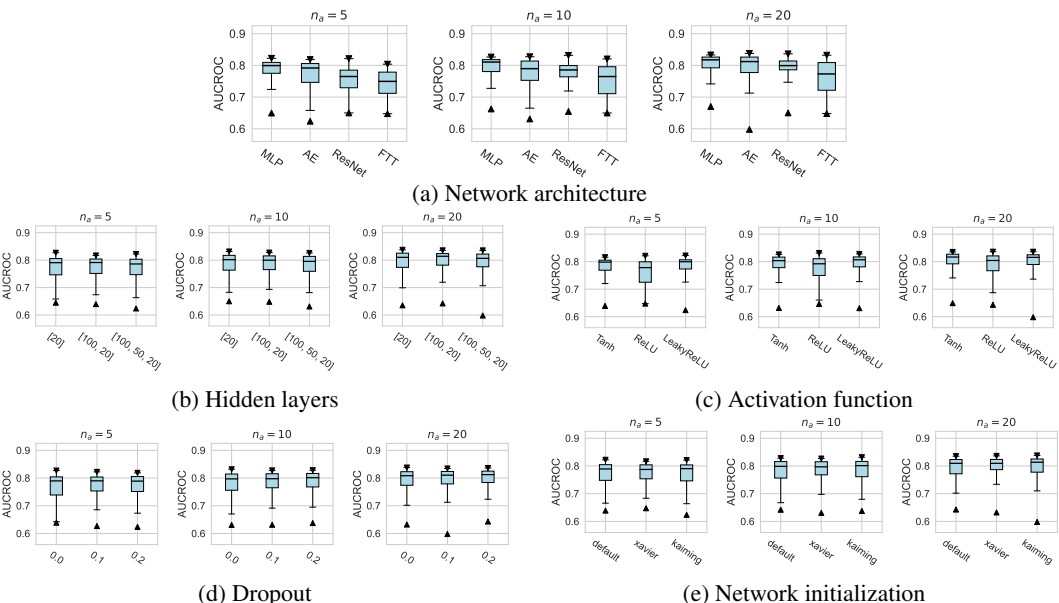

Figure 3: AUCROC performance of network construction designs. MLP is still an effective architecture for AD tasks, where Tanh and LeakyReLU activation functions are generally better than ReLU.

### 4.2.3 Network Training

In this subsection, we evaluate different design dimensions in the network training step, as shown in Fig. 4. Based on the results, we surprisingly observe that the classical BCE loss could be a competitive baseline when batch resampling method[1] is applied in the training process. Moreover, our results show the advantages of hinge loss [56, 59], which achieves better AUCROC performance w.r.t. different $n_a$, and deviation loss [59] in all synthetic datasets with a significant similarity across the anomalies. (details in Appx. D). For model optimization, we observe that Adam and RMSprop optimizers are better than the classical SGD, where large training epochs (e.g., epochs=100) lead to overfitting on limited labeled data, resulting in a relatively inferior performance. For the design dimensions of batch size, learning rate, and weight decay, they do not show obvious differences in terms of the AUCROC metric.

### 4.3 Automatic Component Construction via ADGym

In this subsection, we verify the effectiveness of the proposed meta-predictors by answering the questions illustrated below. Beyond the fixed SOTA AD methods, we provide three model construction plans for a thorough baseline comparison. The random selection plan (RS) randomly generates a pipeline from all design choices. The supervised selection plan (SS) selects the pipeline according

---

[1]Batch resampling method [59] randomly samples a training batch with half of the samples from the unlabeled data and another half from the labeled anomalies. We applied this method for each loss function in order to mitigate the impact of the class imbalance problem, except for the inverse loss, as it significantly affects the gradient update of the neural network. More descriptions of these methods can be found in the Appx. C.

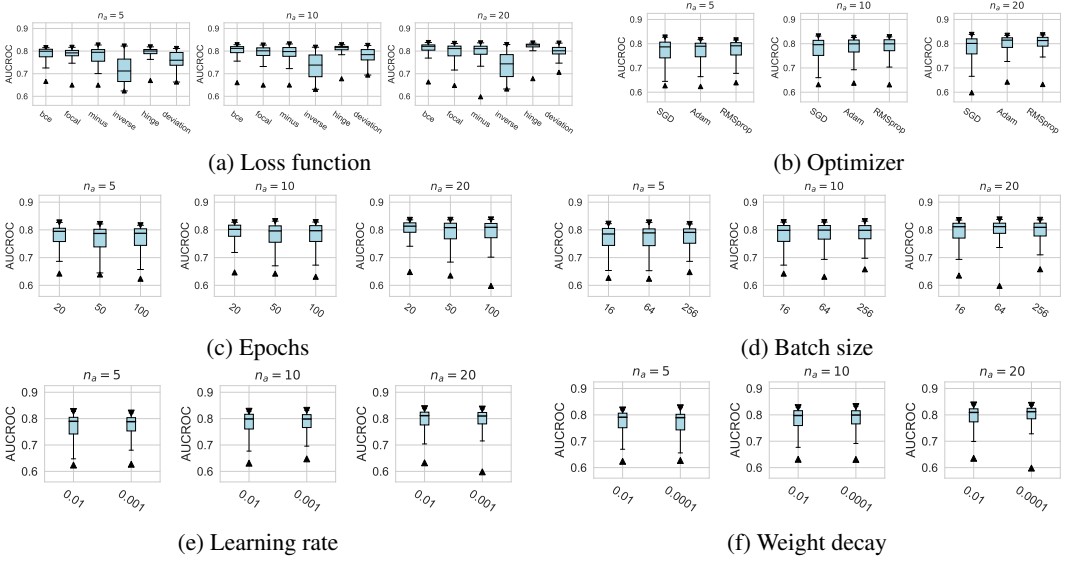

Figure 4: AUCROC performance of network training designs. Hinge loss is generally a better AD loss function. Both Adam and RMSprop optimizers outperform classical SGD. Smaller epochs could prevent overfitting on limited labeled data and thus achieve better performance.

to the evaluation of validation sets using labels. The ground truth plan (GT) is the best-performing pipeline in a sample of 1000. We iteratively leave one dataset as the testing dataset, and leverage all the remaining datasets to train the meta-predictor(s). The central experimental results of ADGym's performance are provided in Table 3. We report the average AUCROC performance across real-world datasets and demonstrate the complete experimental results in the Appx. D.

In the subsequent paragraphs, we present a comparison between the AD models constructed by the meta-predictor and the SOTA AD methods. Also, we discuss the impact of meta-predictors' design on their performance, including meta-features, the loss function of the predictors, and more.

**1. Is the model constructed by meta-predictor better than existing SOTA methods?**

Table 2: Performance of the baseline SOTA AD methods.

| $n_a$ | GANomaly | REPEN | DeepSAD | DevNet | FEAWAD | ResNet | FTTransformer |
|---|---|---|---|---|---|---|---|
| 5 | 0.605 | **0.784** | 0.718 | 0.761 | 0.745 | 0.617 | 0.782 |
| 10 | 0.616 | 0.787 | 0.750 | 0.792 | 0.789 | 0.673 | **0.816** |
| 20 | 0.615 | 0.801 | 0.780 | 0.828 | 0.829 | 0.735 | **0.843** |

Table 3: Performance of auto-selected pipelines by ADGym. RS, SS, and GT refer to the random selection, supervised selection, and the best performance among all sampled 1000 pipelines, respectively. DL or ML corresponds to the meta-predictor that is either instantiated with MLP or tree-based methods like XGBoost. The suffix -ensemble refers to ensemble multiple predictions of meta-predictors. "2-stage" and "end2end" correspond to the two-stage and end-to-end for extracting meta-features.

| $n_a$ | RS | SS | GT | DL-single | | DL-ensemble | | ML-single | | ML-ensemble | |
|---|---|---|---|---|---|---|---|---|---|---|---|
| | | | | 2-stage | end2end | 2-stage | end2end | XGB | CatB | XGB | CatB |
| 5 | 0.735 | 0.613 | 0.902 | 0.800 | 0.811 | 0.813 | 0.813 | 0.802 | 0.804 | 0.815 | **0.821** |
| 10 | 0.757 | 0.696 | 0.912 | 0.836 | 0.833 | 0.843 | 0.837 | 0.836 | 0.839 | **0.849** | 0.847 |
| 20 | 0.777 | 0.747 | 0.921 | 0.859 | 0.850 | 0.865 | 0.857 | 0.857 | 0.861 | **0.869** | 0.872 |

Our results show that the automatic construction of AD models is indeed capable of *surpassing* those SOTA methods, where they achieve better AUCROC performance in all settings, and this conclusion holds true under various numbers of labeled anomalies $n_a$. Specifically, the average/best AUCROC performance of meta-predictors shows a relative improvement over the best SOTA models REPEN 2.6%/3.7% w.r.t. $n_a$=5, FTTransformer 2.4%/3.3% w.r.t. $n_a$=10 and 1.8%/2.9% w.r.t. $n_a$=20.

Moreover, the average AUCROC performance of meta-predictors is 9.4%, 9.4%, and 8.5% higher than that of SOTA models w.r.t. $n_a = 5$, 10, and 20, respectively.

Moreover, the clear advantage of GT to the current SOTA methods indicates that existing AD solutions could be further improved through exploring the design space or automatic selection techniques, which also validates the value of ADGym. Besides, compared to random selection (RS) and supervised selection (SS) which select design choices based on the performance of limited labeled data, automatic selection via meta-predictor(s) is significantly more effective. However, we find that the SS method is even inferior to the RS method. A reasonable explanation is that the scarcity of labeled data makes it difficult for the selected design choices to generalize well on newcoming data.

**2. Is it more helpful for meta-predictors to transfer knowledge across different datasets by end-to-end trained meta-features or extraction-based meta-features?**

Here, we explore the impact of different methods for extracting meta-features on meta-predictor performance, as shown in Table 3. This includes: *(i)* Two-stage method, where the meta-features of a specific dataset are first extracted by the method proposed in MetaOD [88], i.e., extracting both statistical features like min, max, variance, skewness, covariance, etc., and landmarker features that depend on the unsupervised AD algorithms to capture outlying characteristics of a dataset. The extracted meta-features $E^{meta}$ are then concatenated with $n_a$ and $E^{comp}$ as inputs to the meta-predictor. *(ii)* End-to-end method, where the meta-features are directly extracted by the meta-predictor [30] and optimized with the learning process of the downstream task.

Generally, we observe *close* performances between the two-stage and end-to-end methods of extracting meta-features in the meta-predictors. This conclusion holds valid not only for the default MSE loss, but also for the Ranknet loss [11] used in the meta-predictors.

**3. Is the tree-based ensemble meta-predictor effective in tabular AD? Can meta-predictors further benefit from model ensembling?**

Consistent with the findings in the previous studies [24, 25, 26], we *still* find that the tree-based ensemble model(s) are better solutions for tabular-based AD tasks, where the performance of meta-predictors are improved when the MLP trainer used in meta-predictor is replaced by the ensemble models like XGBoost and CatBoost. Moreover, DL-based meta-predictor can also benefit from the model ensembling strategy, as we observe that both two-stage and end-to-end meta-predictors improve when we ensemble the anomaly scores of predicted top-$k$ combinations of design choices.

**4. Do advanced loss functions bring performance gains to the meta-predictor?**

In addition to the default MSE loss function, we also investigate several other losses (as shown in Table 4), including *(i)* Weighted MSE imposes a stronger penalty on errors in predicting the top and bottom design choices. *(ii)* Pearson correlation between the predictions and ground-truth targets. *(iii)* Ranknet [11] learns to rank different design choices, which is widely used in recommendation systems. However, compared to the results of MSE loss shown in Table 3, we *do not* find significant improvement when more complicated loss functions are implemented to train the meta-predictors.

Table 4: AUCROC performance of meta-predictors trained on other loss functions. Different loss functions do not yield significant performance improvement for the meta-predictor.

| $n_a$ | Weighted MSE | | | | Pearson | | | | Ranknet | | | |
|---|---|---|---|---|---|---|---|---|---|---|---|---|
| | DL-single | | DL-ensemble | | DL-single | | DL-ensemble | | DL-single | | DL-ensemble | |
| | 2-stage | end2end | 2-stage | end2end | 2-stage | end2end | 2-stage | end2end | 2-stage | end2end | 2-stage | end2end |
| 5 | 0.801 | 0.748 | 0.811 | 0.774 | 0.778 | 0.808 | 0.804 | 0.815 | 0.800 | 0.809 | **0.821** | 0.813 |
| 10 | 0.830 | 0.770 | 0.842 | 0.786 | 0.825 | 0.833 | 0.834 | 0.843 | 0.835 | 0.836 | 0.841 | **0.844** |
| 20 | 0.844 | 0.786 | **0.862** | 0.801 | 0.839 | 0.840 | 0.852 | 0.854 | 0.853 | 0.853 | **0.862** | 0.859 |

**5. Does larger AD design space bring performance gains to the meta-predictors?**

In order to explore whether the meta-predictors can uncover more promising AD design choices beyond those bolded design dimensions highlighted in Table 1, we include more possible design dimensions like hidden layers, dropout, and initialization methods in network construction, and epochs, batch size, and weight decay in network training, while maintaining the scale of design space (i.e., the number of Cartesian products of design choices) at 1,000. Table 5 shows that the meta-predictors generally benefit from a larger design space, where the AUCROC performances of both DL- and ML-based meta-predictors achieve improvements compared to that of small design space

shown in Table 3. It's worth mentioning that DL meta-predictors seem to surpass ML meta-predictors in a larger space. This may be due to a raised complexity and lower probability of over-fitting in a larger selection pool, both benefiting DL meta-predictors. However, this performance gap is still trivial (0.002-0.006 in absolute value). Considering a better efficiency, we still present the ML meta-predictors as the formal recommendation.

Table 5: AUCROC performance of meta-predictors trained on large scale design space. Larger design space provides potentially more effective design choices for newcoming datasets.

| $n_a$ | RS | SS | GT | DL-single | | DL-ensemble | | ML-single | | ML-ensemble | |
|---|---|---|---|---|---|---|---|---|---|---|---|
| | | | | 2-stage | end2end | 2-stage | end2end | XGB | CatB | XGB | CatB |
| 5 | 0.738 | 0.657 | 0.904 | 0.824 | 0.808 | **0.829** | 0.826 | 0.814 | 0.814 | 0.825 | 0.825 |
| 10 | 0.767 | 0.731 | 0.912 | 0.842 | 0.830 | **0.853** | 0.850 | 0.843 | 0.846 | 0.850 | 0.851 |
| 20 | 0.791 | 0.750 | 0.922 | 0.863 | 0.846 | **0.876** | 0.859 | 0.860 | 0.863 | 0.870 | 0.874 |

**6. Does refining AD design space bring performance gains to the meta-predictor?**

We further refined the design space in ADGym and excluded the design choices whose average performances on the training datasets are below the median, resulting in better yet fewer design choices that can be utilized for training meta-predictors. The refined results are reported in Table 6, compared to Table 3, We have *not* found that refining the design space brings any gains to the meta-predictor, possibly because such approach loses many potential design choices that could perform well on newcoming dataset, or loses (half of) the training data of meta-predictors.

Table 6: AUCROC performance of meta-predictors trained on refined design space. No significant improvement of meta-predictor is observed since not only potentially better design choices are discarded, but also results in a reduction of training data in meta-predictors.

| $n_a$ | RS | SS | GT | DL-single | | DL-ensemble | | ML-single | | ML-ensemble | |
|---|---|---|---|---|---|---|---|---|---|---|---|
| | | | | 2-stage | end2end | 2-stage | end2end | XGB | CatB | XGB | CatB |
| 5 | 0.766 | 0.653 | 0.902 | 0.797 | 0.809 | 0.812 | 0.817 | 0.798 | 0.807 | 0.816 | **0.819** |
| 10 | 0.803 | 0.700 | 0.912 | 0.824 | 0.833 | 0.838 | 0.835 | 0.838 | 0.830 | **0.848** | 0.842 |
| 20 | 0.833 | 0.754 | 0.921 | 0.851 | 0.830 | 0.867 | 0.848 | 0.874 | 0.867 | **0.877** | 0.872 |

## 5 Conclusions, Limitations, and Future Directions

In this paper, we introduce ADGym, a novel platform designed for benchmarking and automating AD design choices. We break down AD algorithms into granular components and systematically assess each one's effectiveness through extensive experiments. Furthermore, we develop an automated method for selecting optimal design choices, enabling the creation of AD models that surpass current SOTA algorithms. Our results highlight the crucial role design choices play and offer a structured approach to optimizing them in model development. With the broader AD community in mind, we have made ADGym openly available. We believe its comprehensive design choice evaluation capabilities will significantly contribute to future advancements in automated AD model generation.

Looking ahead, we see several opportunities to enhance ADGym and broaden its application. Firstly, we plan to incorporate time-series AD tasks that handle data with temporal variations. By automating the design choice selection, AD models will be better equipped to adapt to these changing distributions, thereby improving anomaly detection in dynamic time-series contexts. Currently, ADGym is geared towards weakly supervised AD. In the future, we aim to include unsupervised neural network-based AD algorithms as well. Additionally, it is worth noting that non-neural-network-based techniques, such as ensemble-tree methods, have shown great promise in practical scenarios. Therefore, exploring automatic pipeline formulation in these areas is an exciting and valuable direction for future research.

## 6 Acknowledgement

We thank anonymous reviewers for their insightful feedback and comments. M.J., C.H., A.Z., S.H., and H.H. are supported by the National Natural Science Foundation of China under Grant No. 72271151, and the National Key Research and Development Program of China under Grant No. 2022YFC3303301. M.J., C.H., A.Z., S.H., and H.H. thank the financial support provided by FlagInfo-SHUFE Joint Laboratory. X.H. and Y.Z. are independently supported by CMU.

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

# Appendix

We provide more details of the evaluated 29 datasets (§A), compared baselines (§B), the proposed ADGym (§C), and additional experimental results (§D).

# A  Dataset List

Most of the datasets used for model evaluation are derived from our previous work [26], and we drop those datasets smaller than 1,000, and use the subsets of 3,000 for those datasets greater than 3,000 due to the computational cost. We also remove those datasets that cause errors for the compared baseline models. This results in a total of 29 datasets, as is described in Table A1. These datasets cover many application domains, including healthcare (e.g., disease diagnosis), audio and language processing (e.g., speech recognition), image processing (e.g., object identification), finance (e.g., financial fraud detection), and more, where we show this information in the last column.

Table A1: Data description of the 29 datasets included in ADGym.

| Data | # Samples | # Features | # Anomaly | % Anomaly | Category | Reference |
|------|-----------|-----------|-----------|-----------|----------|-----------|
| ALOI | 49534 | 27 | 1508 | 3.04 | Image | [19] |
| annthyroid | 7200 | 6 | 534 | 7.42 | Healthcare | [61] |
| backdoor | 95329 | 196 | 2329 | 2.44 | Network | [55] |
| campaign | 41188 | 62 | 4640 | 11.27 | Finance | [59] |
| Cardiotocography | 2114 | 21 | 466 | 22.04 | Healthcare | [7] |
| celeba | 202599 | 39 | 4547 | 2.24 | Image | [59] |
| census | 299285 | 500 | 18568 | 6.20 | Sociology | [59] |
| donors | 619326 | 10 | 36710 | 5.93 | Sociology | [59] |
| fault | 1941 | 27 | 673 | 34.67 | Physical | [19] |
| landsat | 6435 | 36 | 1333 | 20.71 | Astronautics | [19] |
| letter | 1600 | 32 | 100 | 6.25 | Image | [21] |
| magic.gamma | 19020 | 10 | 6688 | 35.16 | Physical | [19] |
| mammography | 11183 | 6 | 260 | 2.32 | Healthcare | [77] |
| mnist | 7603 | 100 | 700 | 9.21 | Image | [37] |
| musk | 3062 | 166 | 97 | 3.17 | Chemistry | [16] |
| optdigits | 5216 | 64 | 150 | 2.88 | Image | [5] |
| PageBlocks | 5393 | 10 | 510 | 9.46 | Document | [54] |
| pendigits | 6870 | 16 | 156 | 2.27 | Image | [4] |
| satellite | 6435 | 36 | 2036 | 31.64 | Astronautics | [64] |
| satimage-2 | 5803 | 36 | 71 | 1.22 | Astronautics | [64] |
| shuttle | 49097 | 9 | 3511 | 7.15 | Astronautics | [64] |
| skin | 245057 | 3 | 50859 | 20.75 | Image | [19] |
| SpamBase | 4207 | 57 | 1679 | 39.91 | Document | [12] |
| speech | 3686 | 400 | 61 | 1.65 | Linguistics | [10] |
| thyroid | 3772 | 6 | 93 | 2.47 | Healthcare | [62] |
| vowels | 1456 | 12 | 50 | 3.43 | Linguistics | [33] |
| Waveform | 3443 | 21 | 100 | 2.90 | Physics | [50] |
| Wilt | 4819 | 5 | 257 | 5.33 | Botany | [12] |
| yeast | 1484 | 8 | 507 | 34.16 | Biology | [28] |
| local | | | | | synthesis | [26] |
| global | | | | | synthesis | [26] |
| cluster | | | | | synthesis | [26] |
| dependency | | | | | synthesis | [26] |

# B  Compared Baselines

We compare the automatically selected AD models via ADGym with the following weakly- and fully-supervised baselines, which are considered as effective AD solutions in the previous work [26].

1. **Semi-Supervised Anomaly Detection via Adversarial Training (GANomaly)** [3]. A GAN-based method defines the reconstruction error of the input data as the anomaly score. We replace the convolutional layer in its original version with the dense layer for the tabular AD task, where the hidden size of the encoder-decoder-encoder structure of GANomaly is set to half of the input dimension. We train the GANomaly for 50 epochs with 64 batch size, where the SGD [66] optimizer with 0.01 learning rate and 0.7 momentum is applied for both the generator and discriminator.

2. **REPresentations for a random nEarest Neighbor distance-based method (REPEN)** [56]. A neural network-based model that leverages transformed low-dimensional embedding for random distance-based detectors. The hidden size of REPEN is set to 20, and the margin of triplet loss is set to 1000. REPEN is trained for 1000 epochs with early stopping. The batch size is set to 256, where the total number of steps (batches of samples) is set to 50. Adadelta [82] optimizer with 0.001 learning rate and 0.95 $\rho$ is applied to update network parameters.

3. **Deep Semi-supervised Anomaly Detection (DeepSAD)** [69]. DeepSAD is a deep one-class method that improves its unsupervised version DeepSVDD [68] by penalizing the inverse distances of anomaly representation such that anomalies must be mapped further away from the hypersphere center. The hyperparameter $\eta$ in the loss function is set to 1.0, where DeepSAD is trained for 50 epochs with 128 batch size. Adam optimizer with 0.001 learning rate and $10^{-6}$ weight decay is applied for updating the network parameters. DeepSAD additionally employs an autoencoder for calculating the initial center of the hypersphere, where the autoencoder is trained for 100 epochs with 128 batch size, and optimized by Adam optimizer with learning rate 0.001 and $10^{-6}$ weight decay.

4. **Deviation Networks (DevNet)** [59]. A neural network-based model uses a prior Gaussian distribution to enforce a statistical deviation score of input instances. The margin hyperparameter $a$ in the deviation loss is set to 5. DevNet is trained for 50 epochs with 512 batch size, where the total number of steps is set to 20. RMSprop [66] optimizer with 0.001 learning rate and 0.95 $\rho$ is applied to update network parameters.

5. **Feature Encoding With Autoencoders for Weakly Supervised Anomaly Detection (FEAWAD)** [90]. A neural network-based model that integrates the network architecture of DAGMM [91] with the deviation loss in DevNet [59]. FEAWAD is trained for 30 epochs with 512 batch size, where the total number of steps is set to 20. Adam optimizer with 0.0001 learning rate is applied to update network parameters.

6. **Residual Nets (ResNet)** [24]. This method introduces a ResNet-like architecture [27] for tabular data. ResNet is trained for 100 epochs with 64 batch size. AdamW [51] optimizer with 0.001 learning rate is applied to update network parameters.

7. **Feature Tokenizer + Transformer (FTTransformer)** [24]. FTTransformer is an effective adaptation of the Transformer architecture [75] for tabular data. FTTransformer is trained for 100 epochs with 64 batch size. AdamW optimizer with 0.0001 learning rate and $10^{-5}$ weight decay is applied to update network parameters.

# C   Details of ADGym

In this section, we provide more details of the design space in ADGym, including the pipelines of Data Handling, Network Construction, and Network Training. Besides, we also introduce detailed descriptions of the extracted meta-features and the trained meta-predictors.

## C.1   Design Choices Specification

**Data Handling**

Data augmentation aims to enhance the quality of training data by generating synthetic anomalies, mitigating the class-imbalance problem in AD tasks. For a dataset $\mathcal{D} = \{\boldsymbol{x}_1^u, \ldots, \boldsymbol{x}_k^u, (\boldsymbol{x}_{k+1}^a, y_{k+1}^a), \ldots, (\boldsymbol{x}_{k+m}^a, y_{k+m}^a)\}$ defined in the main paper, the output after a specific data augmentation method is $D' = \{\boldsymbol{x_1^u}, ..., \boldsymbol{x_k^u}, (\boldsymbol{x_{k+1}^a}, y_{k+1}^a), ..., (\boldsymbol{x_{k+m}^a}, y_{k+m}^a), (\boldsymbol{x_{k+m+1}^a}, y_{k+m+1}^a), ..., (\boldsymbol{x_{k+m+j}^a}, y_{k+m+j}^a)\}$, where $m \ll k$ and $m + j \approx k$. This will exhibit a more balanced distribution between the normal and abnormal ones. Except for the default setting (maintaining the original class distribution of the dataset), we present four additional choices in this dimension: Oversampling, SMOTE [14], Mixup [39, 74, 84], and GAN [6].

Data preprocessing includes two commonly used methods, MinMaxScaler and Normalization. Given the input data $D \in \mathbb{R}^{n \times d}$, for each feature vector $F \in \mathbb{R}^n$, MinMax scales $F$ to the range $[a, b]$: $F' = (F - F_{min}) * (b - a)/(F_{max} - F_{min}) + a$, where $F_{min}$ and $F_{max}$ represent the minimum and maximum of $F$. For each sample vector $S \in \mathbb{R}^d$, Normalization compute its $L2$ norm and scale $S$ to $S' = S/||S||^2$ if $||S|| \neq 0$.

**Network Construction**

Network Construction is often regarded as an important part in the CV or NLP domain, while it is often neglected in the AD problem. In fact, some design choices like AutoEncoder [90] and Transformer [24] could facilitate the detection performance of downstream tabular AD tasks. We comprehensively present the entire network construction process from various aspects, such as network architecture, hidden layers, activation functions, dropout rate, and parameter initialization, and provide numerous design choices.

**Network Training**

Many researchers regard the loss function as the key component of AD models. Binary cross entropy (BCE) loss is the conventionally accepted choice for classification tasks but is often considered inappropriate for AD. However, its value as a robust baseline may be overlooked by the AD community, as we have pointed out in the

main part §4.2.3. In ADGym, we explore the potential of diverse loss functions under different combinations of design choices.

As an overview of these loss functions, Focal loss [45] increases the weighting of abnormal data in cross-entropy loss, enabling the model to focus more on the classification of anomalies. Minus loss [18] offers divergent update directions for anomaly scores amidst normal and abnormal data, and circumvents the issue of loss explosion with an upper bound. Resembling minus loss in approach, Inverse loss [69] is inherently self-limiting. Hinge loss [56, 59] leverages a pre-established hyperparameter, $M$, to maintain an anomaly score margin of a minimum of $M$. Deviation loss [59] mandates that the anomaly scores of normal data align with a standard Gaussian distribution, simultaneously ensuring a minimal $M$ margin for anomaly scores. Other dimensions here are consistent with common DL.

## C.2 Meta-features and Meta-predictors

**Details and the selected list of meta-features**

Data characterization measures which represent task information are fundamental for meta-learning [65]. We extract multiple categories of them as meta-features in the reference of [88], including: (1) *Statistical* features depict the dataset from multiple statistical indicators, including but not limited to characteristics such as the minimum value, maximum value, mean, and variance, as well as combinations of these features. These characteristics are widely used in the previous AutoML studies, demonstrating their feasibility for application. (2) *Landmarker* features are constructed by a series of AD models, including iForest, HBOS, LODA, and PCA (anomalies are reconstructed harder), which represent more AD task-specific information. For each of the four different unsupervised AD models trained by a given dataset, we extract their characteristics to engineer a series of meta-features (See Table C3 for complete landmaker features). Overall, the experiment on the proposed two-stage method has proven that we can construct similar models for datasets with similar landmarker meta-features.

Table C2: Selected meta-features for characterizing an arbitrary dataset.

| Name | Formula | Rationale | Variants |
|---|---|---|---|
| Nr instances | n | Speed, Scalability | $\frac{p}{n}$, $\log(n)$, $\log(\frac{n}{p})$ |
| Nr features | p | Curse of dimensionality | $\log(p)$, % categorical |
| Sample mean | $\mu$ | Concentration | |
| Sample median | $\tilde{X}$ | Concentration | |
| Sample var | $\sigma^2$ | Dispersion | |
| Sample min | $\max_X$ | Data range | |
| Sample max | $\min_X$ | Data range | |
| Sample std | $\sigma$ | Dispersion | |
| Percentile | $P_i$ | Dispersion | q1, q25, q75, q99 |
| Interquartile Range (IQR) | $q75 - q25$ | Dispersion | |
| Normalized mean | $\frac{\mu}{\max_X}$ | Data range | |
| Normalized median | $\frac{\tilde{X}}{\max_X}$ | Data range | |
| Sample range | $\max_X - \min_X$ | Data range | |
| Sample Gini | | Dispersion | |
| Median absolute deviation | $\text{median}(X - \tilde{X})$ | Variability and dispersion | |
| Average absolute deviation | $\text{avg}(X - \tilde{X})$ | Variability and dispersion | |
| Quantile Coefficient Dispersion | $\frac{(q75-q25)}{(q75+q25)}$ | Dispersion | |
| Coefficient of variance | | Dispersion | |
| Outlier outside 1 & 99 | % samples outside 1% or 99% | Basic outlying patterns | |
| Outlier 3 STD | % samples outside $3\sigma$ | Basic outlying patterns | |
| Normal test | If a sample differs from a normal dist. | Feature normality | |
| $k$th moments | | | 5th to 10th moments |
| Skewness | Feature skewness | Feature normality | min, max, $\mu$, $\sigma$, skewness, kurtosis |
| Kurtosis | $\frac{\mu_4}{\sigma^4}$ | Feature normality | min, max, $\mu$, $\sigma$, skewness, kurtosis |
| Correlation | $\rho$ | Feature interdependence | min, max, $\mu$, $\sigma$, skewness, kurtosis |
| Covariance | Cov | Feature interdependence | min, max, $\mu$, $\sigma$, skewness, kurtosis |
| Sparsity | $\frac{\#\text{Unique values}}{n}$ | Degree of discreteness | min, max, $\mu$, $\sigma$, skewness, kurtosis |
| ANOVA p-value | $p_{\text{ANOVA}}$ | Feature redundancy | min, max, $\mu$, $\sigma$, skewness, kurtosis |
| Coeff of variation | $\frac{\sigma_x}{\mu_x}$ | Dispersion | |
| Norm. entropy | $\frac{H(X)}{\log_2 n}$ | Feature informativeness | min,max, $\sigma$, $\mu$ |
| Landmarker (HBOS) | See Table C3 | Outlying patterns | Histogram density |
| Landmarker (LODA) | See Table C3 | Outlying patterns | Histogram density |
| Landmarker (PCA) | See Table C3 | Outlying patterns | Explained variance ratio, singular values |
| Landmarker (iForest) | See See Table C3 | Outlying patterns | # of leaves, tree depth, feature importance |

**Detailed and Complete List of Meta-predictors** We represent each design choice with LabelEncoder class from the scikit-learn library. These encoded data, together with the meta-features and the number of labeled anomalies $n_a$, are considered as the input data of the following three kinds of meta-predictors.

**Two-stage meta-predictor**. This meta-predictor uses the meta-features described above. It firstly extracts meta-features $E_i^{meta}$ from the given training datasets as an essential input, and then combines these meta-features with design choice embeddings and $n_a$, learning to predict the relative performance rank.

Table C3: Landmarker features. An example of feature is the minimization of tree depth from the model iForest.

| Model | Perspective | Variants |
|---|---|---|
| iForest | Tree depth
Number of leaves
Mean of base tree feature importance
Max of base tess feature importance | min,max,mean,std,skewness,kurtosis |
| HBOS | Mean of each histogram(per feature)
Max of each histogram(per feature) | min,max,mean,std,skewness,kurtosis |
| LODA | Mean of each random projection(per feature)
Max of each random projection(per feature)
Mean of each histogram(per feature)
Max of each histogram(per feature) | min,max,mean,std,skewness,kurtosis |
| PCA | Explained variance ratio on the first three principal components | The percentage of variance it captures for the top 3 principal components |
| | Singular values | The top 3 singular values generated during SVD process |

**End-to-end meta-predictor**. This meta-predictor learns meta-features implicitly through an end-to-end fashion that is inspired by Dataset2Vec[30].

**Ensembled version** of the above two types of meta-predictors. It has been proved that ensemble models, such as XGBoost and CatBoost, perform well in AD tasks, which motivates us to explore whether the ensemble meta-predictor exhibits superior performance in experiments. From another perspective, this is also a validation of whether our proposed meta-predictor possesses high robustness. Specifically speaking, for each AD task (dataset), we select the top-$k$ (default $k = 5$ in our experiments) best-performance design pipelines predicted by meta-predictor, and utilize the average voting strategy to output the anomaly scores.

**Details on training strategies of meta-predictors**

We experimented with different loss functions to better guide the meta-predictor in selecting the top-$k$ design pipelines. During the training process of meta-predictors, weighted MSE loss assigns greater weights for those design choices with both higher real and predicted performance. The former aims to help the model focus on accurately predicting high-performing design pipelines, while the latter avoids mistakenly predicting poor-performing pipelines as part of the top-$k$. Pearson loss emphasizes the linear correlation between the predicted performances of different design choices and their actual ones. Ranknet loss [11], a commonly used pairwise algorithm in Learning to Rank, aims to model the probability that one design pipeline is superior to another in a probabilistic manner.

# D  Additional Experimental Results

## D.1  Additional Results of Large Evaluations on AD Design Choices

### D.1.1  Using AUCPR as Metrics

For AUCPR performances, we evaluate different design dimensions via the box plots, as shown in Figure D1, D2, and D3, respectively. Overall, we find that the conclusions drawn based on either AUCROC (shown in the main paper) or AUCPR are consistent.

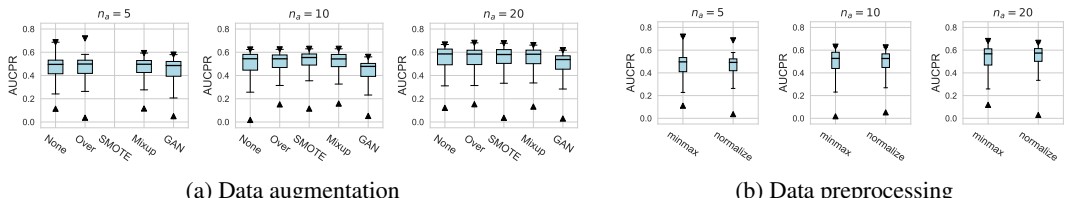

(a) Data augmentation  (b) Data preprocessing

Figure D1: AUCPR performance of data handling designs. There is no significant difference between different design choices in both data augmentation and preprocessing dimensions.

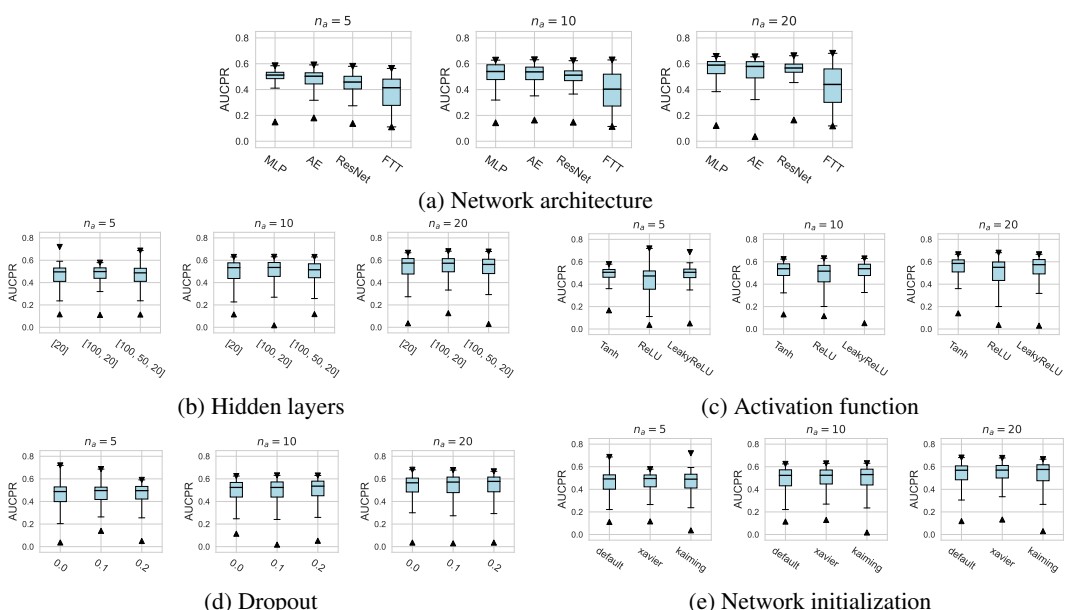

Figure D2: AUCPR performance of network construction designs. MLP is still an effective architecture for AD tasks, where Tanh and LeakyReLU activation functions are generally better than ReLU.

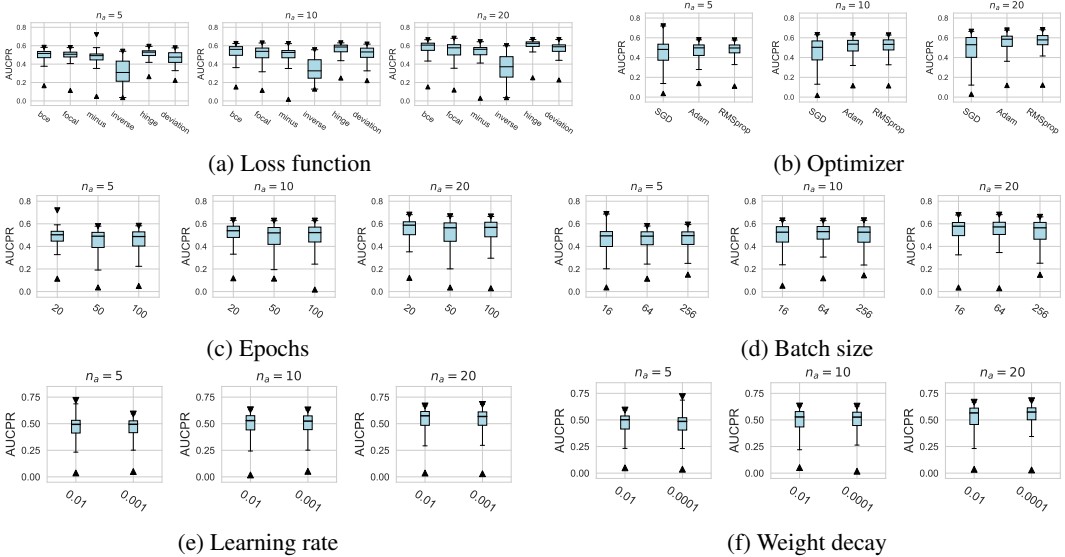

Figure D3: AUCPR performance of network training designs. Hinge loss is generally a better AD loss function. Both Adam and RMSprop optimizers outperform classical SGD. Smaller epochs could prevent overfitting on limited labeled data and thus achieve better performance.

### D.1.2 Using Relative Rank as Metrics

Considering the varying degrees of difficulty across different datasets, we also demonstrate the relative rankings of both AUCROC (as shown in figure D4~D6) and AUCPR (as shown in figure D7~D9) to measure the performance of different design choices. Specifically, we rank 1000 sampled design choices on each dataset. For a more intuitive comparison with our previous results, we calculate $1 - normalized(rank(design\ choice))$, i.e., the inverse of the normalized rank value, as the metric, where higher values indicate better performance. We found that such a metric further amplifies the differences between various design choices and generally aligns with the previous experimental findings. Notably, under the metric of relative ranking, we observe a significant decline in the performance of ResNet, which is inferior to AE. Drawing from our previous experiments, it becomes evident that ResNet's stability across various design pipelines did not manifest under this metric, which, however, is a crucial advantage in practical scenarios.

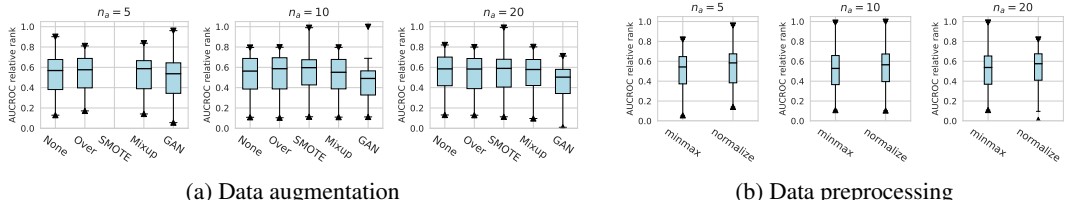

(a) Data augmentation          (b) Data preprocessing

Figure D4: AUCROC relative rank performance of data handling designs.

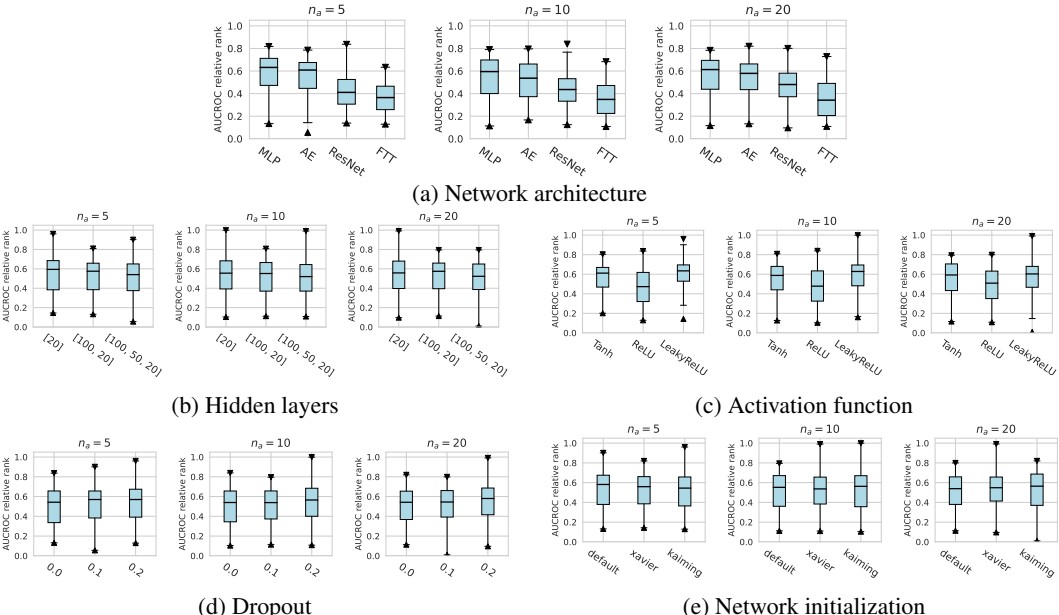

(a) Network architecture

(b) Hidden layers          (c) Activation function

(d) Dropout          (e) Network initialization

Figure D5: AUCROC relative rank performance of network construction designs.

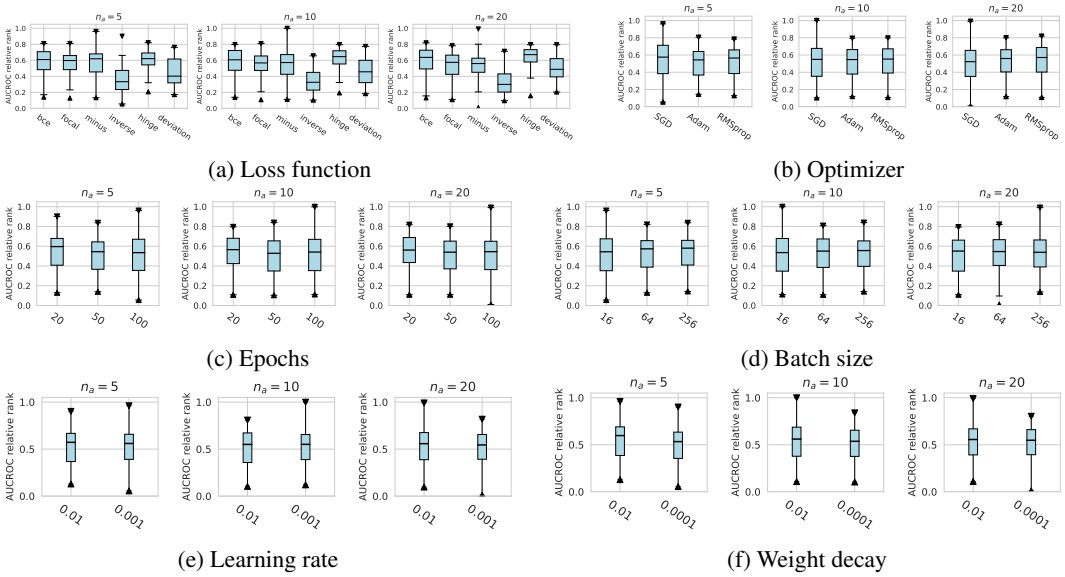

Figure D6: AUCROC relative rank performance of network training designs. Hinge loss is generally a better AD loss function.

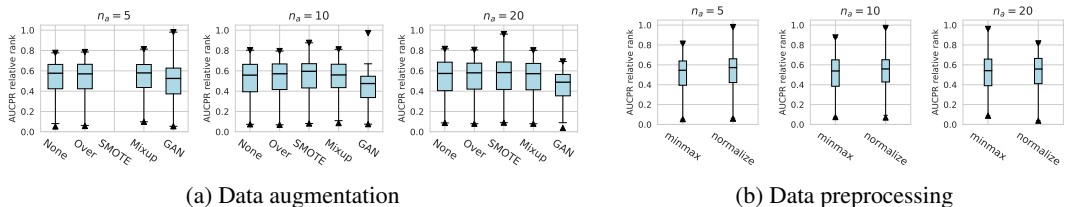

Figure D7: AUCPR relative rank performance of data handling designs.

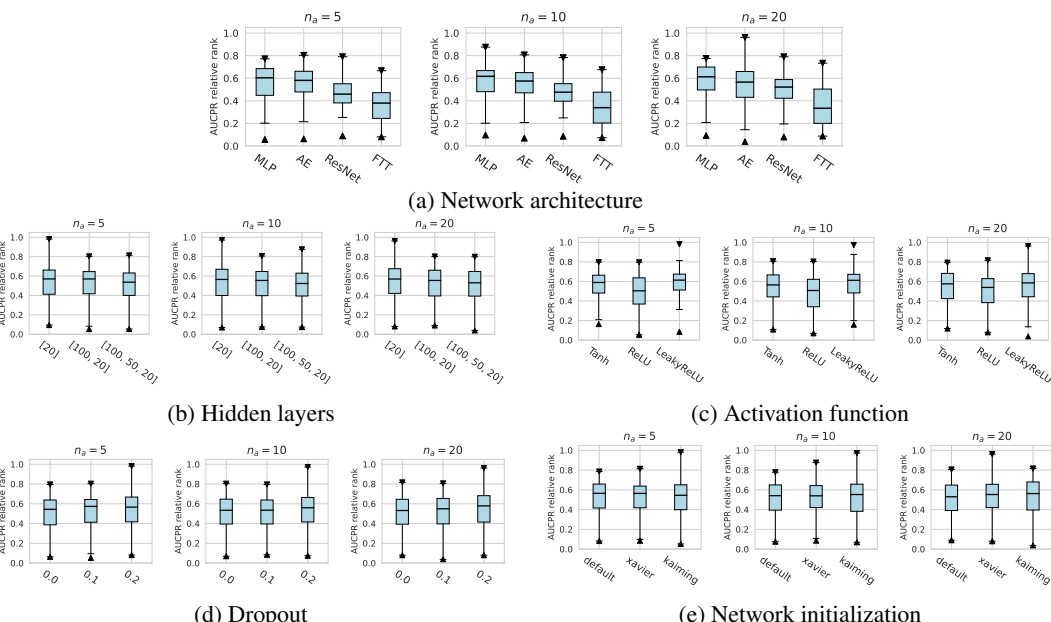

Figure D8: AUCPR relative rank performance of network construction designs.

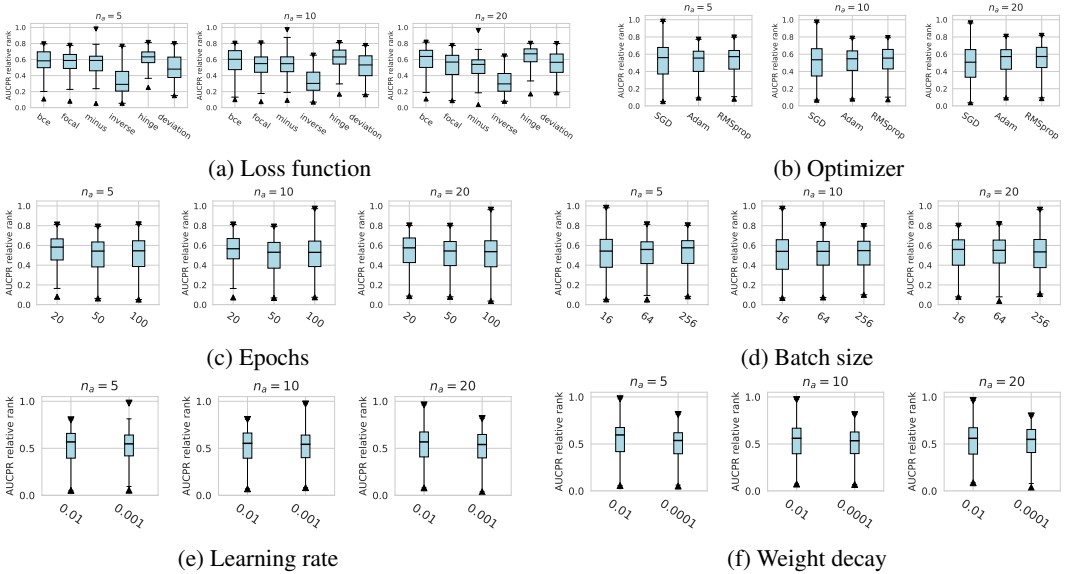

(a) Loss function            (b) Optimizer

(c) Epochs            (d) Batch size

(e) Learning rate            (f) Weight decay

Figure D9: AUCPR relative rank performance of network training designs.

### D.1.3 Unsupervised Network Pre-training

In addition to the existing network initialization methods introduced in Table 1, we also investigate whether deep learning based AD models can benefit from unsupervised pre-training tasks. For tabular AD problems, we follow [68, 69] to first train an AutoEncoder that has an encoder with the same architecture as the corresponding network, e.g., constructing encoder-decoder transformer block in FTTransformer, and then utilize the reconstruction loss (mean squared error between the input data and its reconstructed ones) to perform model training. After that, we initialize corresponding models with the converged parameters of the encoder and fine-tune them on the downstream tasks, which contain only limited labeled data.

Pre-training has been widely used for NLP [15] and CV [52] tasks and verified to significantly enhance the performance of downstream tasks. However, for tabular AD problems, we did not observe a significant advantage of pre-training over other network initialization methods, as shown in Figure D10. This may be due to the reason that compared to the NLP and CV tasks that contain rich textual semantics and visual patterns, the inherent structure of tabular data is more rigid and constrained, resulting in the model struggling to learn general features without label guidance.

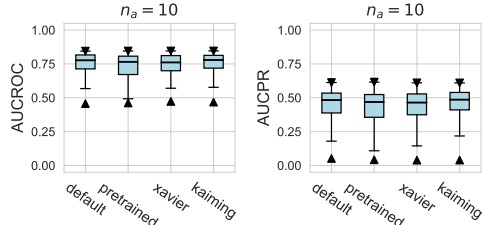

Figure D10: AUCROC and AUCPR performance of network pre-training.

### D.2 Additional Results for Different Anomalies of Synthetic Datasets

Based on 29 real-world scenario datasets, we generated synthetic datasets using three random seeds with four specific anomaly types: local, global, dependency, and cluster. These types were defined in our previous work [26]. After excluding datasets that could not be generated properly, we conducted two experiments on remaining synthetic datasets (totaling over 300): a large benchmark for design choices and a performance experiment for ADGym using global anomaly types as an example.

### D.2.1 Large Evaluation on Design Choices over Synthetic Anomalies

This evaluation is conducted to expose the correlation between the effectiveness of different design choices and the characteristics of the datasets. To more intuitively display our experimental results, we selected three

representative design dimensions: data augmentation, network architecture, and loss function. As shown in figure D11, data augmentation techniques tend to have counterproductive effects on single-anomaly datasets in the vast majority of cases. Figure D12 shows the performance over different network architectures, where ResNet consistently excels over other architectures when dealing with dependency anomalies. However, its performance diminishes with datasets that have a global anomalies context and fewer anomalies. Compared to the two dimensions mentioned above, the variance between different loss functions is significantly greater, making the selection of an appropriate loss function crucial, as shown in figure D13. The deviation loss consistently demonstrates competitive results across various anomaly types and numbers of anomalies, and the performance of hinge loss also improves rapidly as the number of anomalies increases.

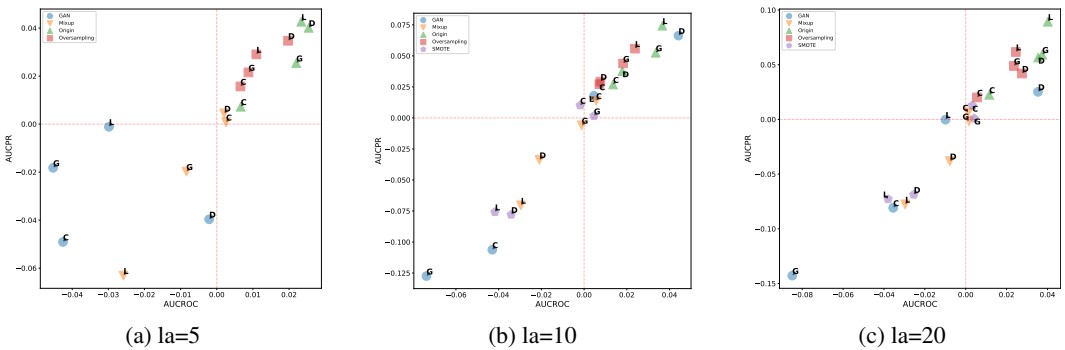

(a) la=5        (b) la=10        (c) la=20

Figure D11: Performance of data augmentation designs across various anomalies. Each point indicates the average performance across three random seeds. Anomalies: global (G), local (L), cluster (C), and dependency (D). The x-axis and y-axis show improvement ratios in AUC-ROC and AUC-PR, respectively

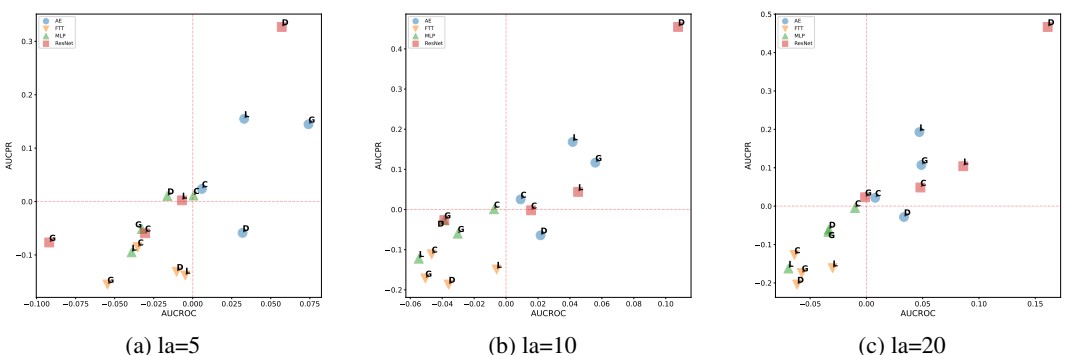

(a) la=5        (b) la=10        (c) la=20

Figure D12: performance of network architecture designs over different type of anomalies.

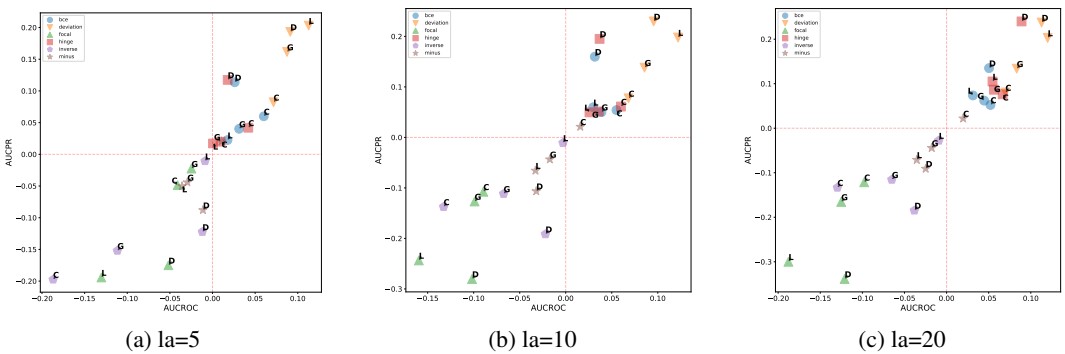

(a) la=5        (b) la=10        (c) la=20

Figure D13: performance of loss function designs over different type of anomalies.

### D.2.2 Additional Results of ADGym over datasets with global anomalies

In this section, we investigate whether meta-predictor can select the best design choices based on dataset meta-features. We chose the meta-predictor trained with two shallow model methods and focused on datasets with global anomalies. These datasets have previously been shown to favor competitive design choices, such as the AE architecture and deviation loss function. Figures D14 and D15 respectively display our results concerning the network architecture and loss function design dimensions, both of which have been shown to significantly impact model performance in earlier experiments. We represent the results using scatter plots, where all results are averaged over three seeds and the number of anomalies; the closer to the top-right corner indicates better performance of the design choice. Empirical evidence shows that the meta-predictor can effectively select the best design choice (AE and deviation loss) in both dimensions, with the prediction results for network architecture almost perfectly aligning with the actual results.

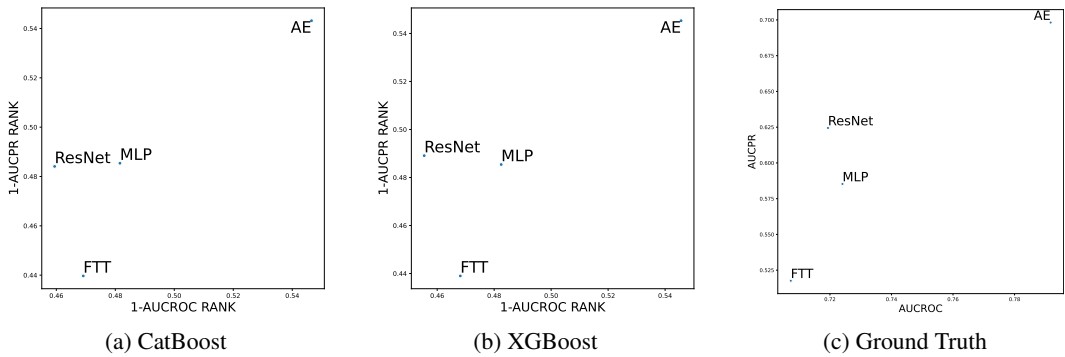

      (a) CatBoost            (b) XGBoost            (c) Ground Truth

Figure D14: performance of meta-predictors on network architecture over global anomalies. Figures (a) and (b) show meta-predictors output trained by CatBoost and XGBoost, and (c) shows result of previous benchmark experiments on global anomalies (as ground truth baseline). For a more intuitive comparison, we subtract the output from 1 to ensure that the best design choices are positioned closer to the top-right corner.

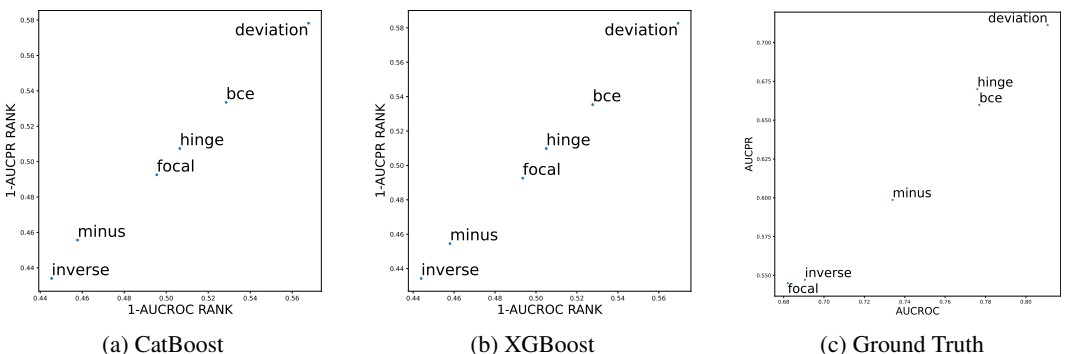

      (a) CatBoost            (b) XGBoost            (c) Ground Truth

Figure D15: performance of meta-predictors on loss function over global anomalies.

### D.3 Additional Results for Different Domains of Datasets

In order to investigate the performance of the AD design choices across different domains, we categorize 29 datasets into 14 distinct domains and aggregate the experimental results based on the domain of each dataset. In Table D4 and D5, each data point represents the average performance rank of the design choices (as indicated by the rows) on the datasets within a specific domain (as indicated by the columns). We observe that no design choices are universally superior or inferior. A case in point is the GAN-based data augmentation method, which generally underperforms on most datasets but shows a clear advantage in the financial domain. A similar situation also occurs with the performance of FTTransformer on datasets in the Sociology domain. Meanwhile, some designs that have performed well in past experiments, such as the deviation loss function, rank last on datasets in the financial domain. These results underscore the necessity of automated selection of AD designs and align with our previous conclusions.

Table D4: AUCROC performance rank of design choices across different domains. First column show different design dimensions, where aug, na, af, ls are abbreviations for data augmentation, network architecture, activate function, and loss function, respectively

| design | method | Astronautics | Biology | Botany | Chemistry | Document | Finance | Healthcare | Image | Linguistics | Network | Physical | Physics | Sociology | Web |
|--------|--------|-----|-----|-----|-----|-----|-----|-----|-----|-----|-----|-----|-----|-----|-----|
| aug | GAN | 5 | 5 | 5 | 5 | 5 | 2 | 5 | 5 | 5 | 5 | 5 | 5 | 4 | 5 |
| aug | Mixup | 2 | 2 | 1 | 3 | 4 | 4 | 4 | 2 | 1 | 4 | 3 | 4 | 2 | 2 |
| aug | Origin | 4 | 3 | 3 | 4 | 3 | 5 | 3 | 4 | 2 | 3 | 2 | 2 | 5 | 3 |
| aug | Oversampling | 3 | 4 | 2 | 2 | 2 | 3 | 2 | 3 | 3 | 2 | 4 | 3 | 3 | 4 |
| aug | SMOTE | 1 | 1 | 4 | 1 | 1 | 1 | 1 | 1 | 4 | 1 | 1 | 1 | 1 | 1 |
| na | AE | 2 | 3 | 2 | 2 | 1 | 2 | 2 | 3 | 4 | 2 | 2 | 2 | 3 | 3 |
| na | FTT | 4 | 4 | 4 | 4 | 4 | 4 | 4 | 4 | 1 | 4 | 4 | 4 | 1 | 4 |
| na | MLP | 1 | 1 | 3 | 1 | 2 | 1 | 1 | 1 | 3 | 1 | 1 | 1 | 2 | 2 |
| na | ResNet | 3 | 2 | 1 | 3 | 3 | 3 | 3 | 2 | 2 | 3 | 3 | 3 | 4 | 1 |
| af | LeakyReLU | 1 | 2 | 1 | 1 | 2 | 1 | 2 | 2 | 3 | 1 | 2 | 1 | 3 | 3 |
| af | ReLU | 3 | 3 | 2 | 3 | 3 | 3 | 3 | 3 | 1 | 3 | 3 | 3 | 2 | 2 |
| af | Tanh | 2 | 1 | 3 | 2 | 1 | 2 | 1 | 1 | 2 | 2 | 1 | 2 | 1 | 1 |
| ls | bce | 3 | 4 | 3 | 1 | 2 | 4 | 4 | 1 | 4 | 1 | 1 | 2 | 1 | 3 |
| ls | deviation | 4 | 3 | 2 | 2 | 4 | 6 | 3 | 3 | 2 | 3 | 3 | 4 | 4 | 1 |
| ls | focal | 5 | 6 | 5 | 4 | 6 | 2 | 5 | 5 | 6 | 5 | 6 | 5 | 5 | 5 |
| ls | hinge | 2 | 1 | 1 | 5 | 3 | 3 | 1 | 2 | 1 | 2 | 4 | 1 | 2 | 2 |
| ls | inverse | 6 | 5 | 6 | 6 | 5 | 5 | 6 | 6 | 5 | 6 | 5 | 6 | 6 | 6 |
| ls | minus | 1 | 2 | 4 | 3 | 1 | 1 | 2 | 4 | 3 | 4 | 2 | 3 | 3 | 4 |

Table D5: AUCPR performance rank of design choices across different domains.

| design | method | Astronautics | Biology | Botany | Chemistry | Document | Finance | Healthcare | Image | Linguistics | Network | Physical | Physics | Sociology | Web |
|--------|--------|-----|-----|-----|-----|-----|-----|-----|-----|-----|-----|-----|-----|-----|-----|
| aug | GAN | 4 | 4 | 4 | 4 | 4 | 1 | 4 | 4 | 4 | 4 | 4 | 4 | 4 | 4 |
| aug | Mixup | 1 | 1 | 1 | 2 | 3 | 3 | 3 | 1 | 1 | 1 | 2 | 1 | 1 | 1 |
| aug | Origin | 3 | 2 | 2 | 3 | 2 | 4 | 2 | 3 | 3 | 3 | 1 | 2 | 3 | 2 |
| aug | Oversampling | 2 | 3 | 3 | 1 | 1 | 2 | 1 | 2 | 2 | 2 | 3 | 3 | 2 | 3 |
| aug | SMOTE | 5 | 5 | 5 | 5 | 5 | 5 | 5 | 5 | 5 | 5 | 5 | 5 | 5 | 5 |
| na | AE | 3 | 3 | 2 | 2 | 1 | 2 | 2 | 3 | 4 | 3 | 2 | 3 | 4 | 3 |
| na | FTT | 4 | 4 | 4 | 4 | 4 | 4 | 4 | 4 | 2 | 4 | 4 | 4 | 1 | 4 |
| na | MLP | 1 | 2 | 3 | 1 | 2 | 1 | 1 | 1 | 3 | 1 | 1 | 1 | 2 | 2 |
| na | ResNet | 2 | 1 | 1 | 3 | 3 | 3 | 3 | 2 | 1 | 2 | 3 | 2 | 3 | 1 |
| af | LeakyReLU | 2 | 3 | 2 | 2 | 2 | 1 | 2 | 2 | 3 | 2 | 2 | 1 | 2 | 2 |
| af | ReLU | 3 | 2 | 1 | 3 | 3 | 3 | 3 | 3 | 1 | 3 | 3 | 3 | 1 | 3 |
| af | Tanh | 1 | 1 | 3 | 1 | 1 | 2 | 1 | 1 | 2 | 1 | 1 | 2 | 3 | 1 |
| lf | bce | 4 | 4 | 3 | 3 | 2 | 5 | 4 | 3 | 3 | 5 | 1 | 2 | 3 | 3 |
| lf | deviation | 1 | 3 | 2 | 1 | 4 | 6 | 2 | 1 | 1 | 3 | 2 | 3 | 2 | 2 |
| lf | focal | 5 | 6 | 5 | 5 | 6 | 2 | 5 | 5 | 6 | 4 | 6 | 5 | 5 | 5 |
| lf | hinge | 2 | 1 | 1 | 4 | 3 | 3 | 1 | 2 | 2 | 2 | 3 | 1 | 1 | 1 |
| lf | inverse | 6 | 5 | 6 | 6 | 5 | 4 | 6 | 6 | 5 | 6 | 5 | 6 | 6 | 6 |
| lf | minus | 3 | 2 | 4 | 2 | 1 | 1 | 3 | 4 | 4 | 1 | 4 | 4 | 4 | 4 |

## D.4 Additional Results of Automatic Component Construction via ADGym

### D.4.1 Using AUCPR as Metrics

In this subsection, we show the AUCPR performance of compared baseline AD methods, as well as different versions of meta-predictors proposed in our ADGym, as is shown in Table D6~D10. We still observe similar results to those illustrated in the main paper, where: (1) the AUCPR performances of automatically constructed AD methods via meta-predictors are better than current state-of-the-art AD solutions. (2) Meta-predictors benefit from either the instantiation of the ensembled tree-based model or an ensembling strategy. (3) Large design space generally improves the performance of meta-predictors.

Table D6: Baseline of SOTA AD methods.

| $n_a$ | GANomaly | REPEN | DeepSAD | DevNet | FEAWAD | ResNet | FTTransformer |
|-------|----------|-------|---------|--------|--------|--------|---------------|
| 5 | 0.235 | 0.480 | 0.333 | 0.484 | 0.470 | 0.363 | 0.489 |
| 10 | 0.240 | 0.476 | 0.417 | 0.530 | 0.530 | 0.433 | 0.561 |
| 20 | 0.250 | 0.517 | 0.491 | 0.584 | 0.579 | 0.528 | 0.623 |

Table D7: Performance of auto-selected pipelines by ADGym. RS, SS, and GT refer to the random selection, supervised selection, and the best performance among all design choices, respectively. DL or ML corresponds to the meta-predictor that is either instantiated with MLP or tree-based method like XGBoost. The suffix -ensemble refers to ensemble multiple predictions of meta-predictors. "2-stage" and "end2end" corresponds to the two-stage and end-to-end for extracting meta-features.

| $n_a$ | RS | SS | GT | DL-single | | DL-ensemble | | ML-single | | ML-ensemble | |
|-------|-----|-----|-----|---------|---------|---------|---------|------|------|------|------|
| | | | | 2-stage | end2end | 2-stage | end2end | XGB | CatB | XGB | CatB |
| 5 | 0.447 | 0.365 | 0.706 | 0.531 | 0.533 | 0.538 | 0.534 | 0.532 | 0.539 | 0.553 | 0.547 |
| 10 | 0.478 | 0.459 | 0.737 | 0.589 | 0.592 | 0.600 | 0.597 | 0.600 | 0.603 | 0.617 | 0.614 |
| 20 | 0.509 | 0.550 | 0.759 | 0.622 | 0.616 | 0.639 | 0.623 | 0.642 | 0.657 | 0.662 | 0.669 |

Table D8: AUCPR performance of meta-predictors trained on other loss functions. Different loss functions do not yield significant performance improvement for the meta-predictor.

| $n_a$ | Weighted MSE | | | | Pearson | | | | Ranknet | | | |
|---|---|---|---|---|---|---|---|---|---|---|---|---|
| | DL-single | | DL-ensemble | | DL-single | | DL-ensemble | | DL-single | | DL-ensemble | |
| | 2-stage | end2end | 2-stage | end2end | 2-stage | end2end | 2-stage | end2end | 2-stage | end2end | 2-stage | end2end |
| 5 | 0.495 | 0.500 | 0.502 | 0.501 | 0.524 | 0.525 | 0.527 | 0.533 | 0.525 | 0.527 | 0.529 | 0.553 |
| 10 | 0.558 | 0.533 | 0.571 | 0.543 | 0.585 | 0.581 | 0.603 | 0.599 | 0.583 | 0.581 | 0.592 | 0.599 |
| 20 | 0.592 | 0.573 | 0.614 | 0.560 | 0.631 | 0.602 | 0.640 | 0.609 | 0.635 | 0.624 | 0.642 | 0.623 |

Table D9: AUCPR performance of meta-predictors trained on large scale design space. Larger design space provides potentially more effective design choices for newcoming datasets.

| $n_a$ | RS | SS | GT | DL-single | | DL-ensemble | | ML-single | | ML-ensemble | |
|---|---|---|---|---|---|---|---|---|---|---|---|
| | | | | 2-stage | end2end | 2-stage | end2end | XGB | CatB | XGB | CatB |
| 5 | 0.449 | 0.373 | 0.716 | 0.539 | 0.537 | 0.547 | 0.556 | 0.554 | 0.544 | 0.558 | 0.551 |
| 10 | 0.491 | 0.455 | 0.742 | 0.590 | 0.583 | 0.613 | 0.604 | 0.593 | 0.588 | 0.609 | 0.622 |
| 20 | 0.528 | 0.557 | 0.762 | 0.641 | 0.617 | 0.656 | 0.630 | 0.624 | 0.641 | 0.660 | 0.662 |

Table D10: AUCPR performance of meta-predictors trained on refined design space. No significant improvement of meta-predictor is observed since not only potentially better design choices are discarded, but also results in a reduction of training data in meta-predictors.

| $n_a$ | RS | SS | GT | DL-single | | DL-ensemble | | ML-single | | ML-ensemble | |
|---|---|---|---|---|---|---|---|---|---|---|---|
| | | | | 2-stage | end2end | 2-stage | end2end | XGB | CatB | XGB | CatB |
| 5 | 0.467 | 0.370 | 0.706 | 0.528 | 0.509 | 0.531 | 0.523 | 0.522 | 0.536 | 0.541 | 0.540 |
| 10 | 0.532 | 0.458 | 0.737 | 0.573 | 0.550 | 0.593 | 0.563 | 0.597 | 0.612 | 0.607 | 0.620 |
| 20 | 0.564 | 0.540 | 0.759 | 0.614 | 0.580 | 0.656 | 0.582 | 0.635 | 0.647 | 0.659 | 0.671 |

### D.4.2 Using Relative Rank as Metrics

In this subsection, we present results using the relative rankings based on AUCROC and AUCPR as metrics to mitigate the impact of variations in dataset difficulty. Specifically, we compute the relative rankings for each datasets across all models, including design pipelines generated by various versions of ADGym meta-predictor. We then subtract these normalized results from 1, ensuring that larger values indicate better performance. Table D11 and Table D12 depict the relative rankings of various SOTA models for AUCROC and AUCPR, respectively, while Table D13 and Table D14 compare the performance of different meta-predictor versions. It's important to note that performance cannot be compared across tables using this relative ranking metric, so we include the average result of the SOTA models as a baseline for ranking purposes. Our findings are consistent with previous conclusions, indicating that the performance of design pipelines from the meta predictor surpasses that of the existing SOTA models.

Table D11: Baseline of SOTA AD methods with AUCROC relative rank.

| $n_a$ | GANomaly | REPEN | DeepSAD | DevNet | FEAWAD | ResNet | FTTransformer |
|---|---|---|---|---|---|---|---|
| 5 | 0.383 | **0.738** | 0.557 | 0.665 | 0.609 | 0.349 | 0.663 |
| 10 | 0.331 | 0.662 | 0.575 | 0.685 | 0.641 | 0.375 | **0.699** |
| 20 | 0.301 | 0.612 | 0.577 | 0.672 | 0.676 | 0.394 | **0.736** |

Table D12: Baseline of SOTA AD methods with AUCPR relative rank.

| $n_a$ | GANomaly | REPEN | DeepSAD | DevNet | FEAWAD | ResNet | FTTransformer |
|---|---|---|---|---|---|---|---|
| 5 | 0.370 | 0.672 | 0.450 | **0.690** | 0.661 | 0.474 | 0.652 |
| 10 | 0.310 | 0.605 | 0.490 | **0.710** | 0.686 | 0.484 | 0.686 |
| 20 | 0.271 | 0.586 | 0.503 | 0.691 | 0.698 | 0.503 | **0.719** |

Table D13: Relative Rank AUCROC of auto-selected pipelines by ADGym.

| $n_a$ | SOTA | RS | SS | GT | DL-single | | DL-ensemble | | ML-single | | ML-ensemble | |
| | | | | | 2-stage | end2end | 2-stage | end2end | XGB | CatB | XGB | CatB |
|---|---|---|---|---|---|---|---|---|---|---|---|---|
| 5 | 0.289 | 0.275 | 0.231 | 0.986 | 0.598 | 0.551 | 0.609 | 0.573 | 0.582 | 0.562 | 0.605 | **0.633** |
| 10 | 0.243 | 0.216 | 0.202 | 0.982 | 0.594 | 0.530 | 0.635 | 0.588 | 0.610 | 0.588 | **0.659** | 0.645 |
| 20 | 0.225 | 0.215 | 0.244 | 0.978 | 0.601 | 0.530 | 0.621 | 0.581 | 0.586 | 0.587 | 0.640 | **0.685** |

Table D14: Relative Rank AUCPR of auto-selected pipelines by ADGym.

| $n_a$ | SOTA | RS | SS | GT | DL-single | | DL-ensemble | | ML-single | | ML-ensemble | |
| | | | | | 2-stage | end2end | 2-stage | end2end | XGB | CatB | XGB | CatB |
|---|---|---|---|---|---|---|---|---|---|---|---|---|
| 5 | 0.353 | 0.284 | 0.276 | 0.984 | 0.572 | 0.545 | 0.600 | 0.552 | 0.557 | 0.566 | **0.608** | 0.596 |
| 10 | 0.275 | 0.297 | 0.270 | 0.982 | 0.563 | 0.527 | 0.599 | 0.569 | 0.547 | 0.569 | 0.628 | **0.668** |
| 20 | 0.260 | 0.319 | 0.313 | 0.973 | 0.549 | 0.514 | 0.614 | 0.539 | 0.536 | 0.600 | 0.617 | **0.659** |

## D.5 Additional Results of Evaluations under Real-world Issues

In real-world applications, AD tasks frequently encounter complex scenarios characterized by noisy and corrupted input data. In this section, we assess the performance and robustness of various AD design choices under three prevalent but imperfect real-world conditions. All experiments are conducted within a weakly-supervised framework, wherein only a sparse set of ten anomaly samples are labeled.

- **Duplicated Anomalies.** Abnormal data are likely to recur multiple times due to various factors. These repeated anomalies often impact many density-based AD algorithms. To simulate this scenario, we duplicated each abnormal samples three times in all datasets for our experiments.
- **Irrelevant Features.** In real-world tabular data, there often exist irrelevant features (columns). Even when feature selection techniques such as Random Forests or Logistic Regression are employed, their efficacy in AD applications is not guaranteed. Therefore, assessing the robustness of AD design choices in the presence of such irrelevant features is crucial. To examine this, we add an additional 30% irrelevant features to all datasets by generating uniform noise.
- **Annotation Errors.** Annotation errors are among the most common forms of noise encountered in real-world scenarios, especially in AD tasks where the data is extremely imbalanced. To assess the robustness of the design choices against label noise, we inverted 10% of the labels in each dataset, limiting this modification to the training sets.

We present the experimental results in Figure D16 and Figure D17 employing both AUCROC and AUCPR metrics. It is evident from the figures that the performance of various design choices is significantly impacted by Irrelevant Features and Annotation Errors, manifesting as a noticeable decline in both metrics. The conclusion variances among different design choices are small under the influence of Irrelevant Features, trending towards convergence. Surprisingly, we find that when Duplicated Anomalies are present, the performance of all design choices tends to improve, and the variance also increases, possibly mitigating the issue of data imbalance. What is particularly encouraging in our experiments is that the ResNet architecture not only shows a substantial and stable advantage in scenarios with Duplicated Anomalies but also exhibits less impact in the other two settings. We also conduct experiments with an Irrelevant Features noisy ratio of 0.1 and an Annotation Errors ratio of 0.05, where only a slight mitigation in the impact on design choices, and the conclusions remained consistent with our current findings.

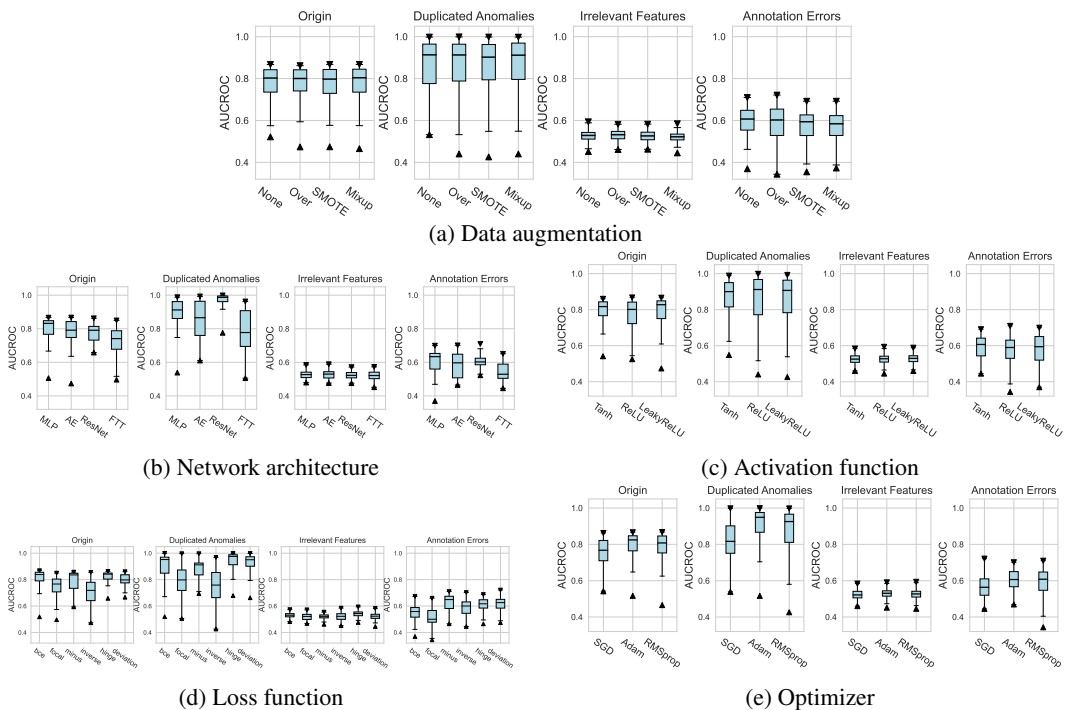

Figure D16: AUCROC performance of design choices under real-world issues, excluding GAN-based methods due to high computational time.

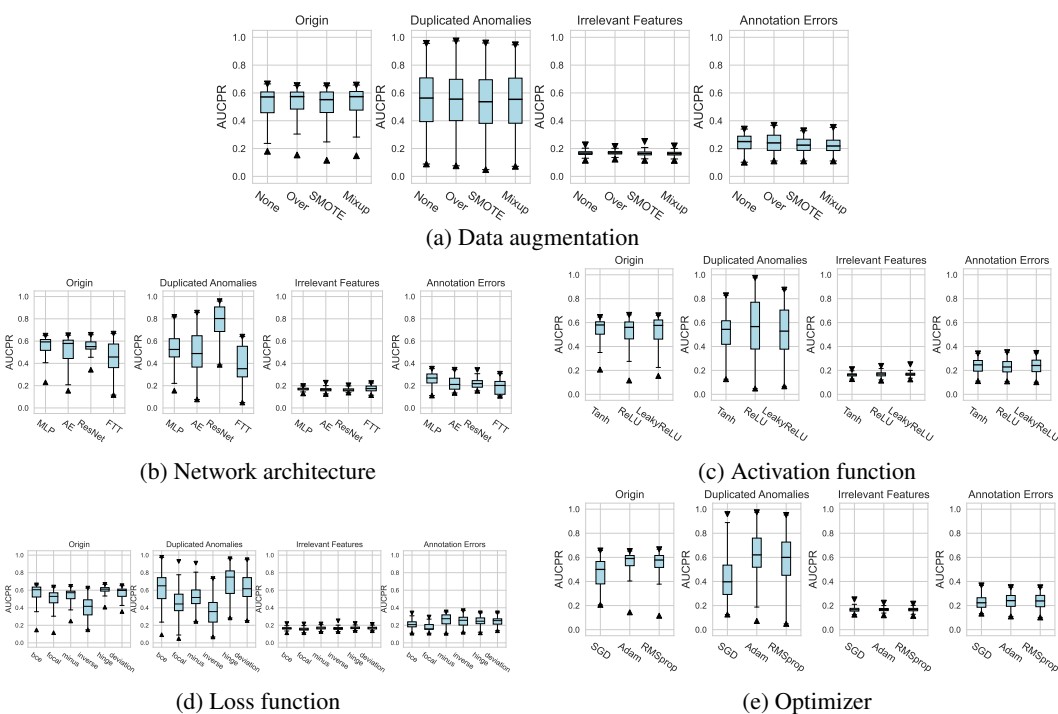

Figure D17: AUCPR performance of design choices under real-world issues.

