# OpenReview forum: "ADGym: Design Choices for Deep Anomaly Detection"
_NeurIPS.cc/2023/Track/Datasets_and_Benchmarks — NeurIPS 2023 Datasets and Benchmarks Poster_

### Official Review · Reviewer_cdKh · 2023-07-21
**Reviews to paper #476**

**Rating:** 6
**Confidence:** 4

**Strengths:**

* The paper proposes a design platform for deep anomaly detection methods, which is the first of its kind. The platform aims to investigate the contribution of each design choice in detecting anomalies, which can help researchers gain a better understanding of the characteristics of deep AD methods. The paper also provides an extensive list of design choices for deep AD models, which can be useful for researchers working in this field.

* Deep anomaly detection is an important research area with numerous applications in finance, medical service, cloud computing, etc. The proposed design platform can help researchers develop more effective AD algorithms by automatically choosing the optimal design choices for a given dataset. The paper's findings can be useful for researchers working in this field and can contribute to the development of more accurate and efficient AD methods.

The paper doesn't need any ethical or social implications.

**Additional Feedback:**

N/A

**Clarity:**

The written of the paper needs to be improved. The paper needs to be carefully revised to make it more clear and easy to follow. Please describe clearly with motivation of the ADGym and its significance in auto-constructing AD models, differentiating from previous HPS/NAS.

**Correctness:**

* The usage of meta-predictor in both WSAD and SupervisedAD is unclear. Line 162-164 described the meta-predictor as regression problem, but not well-stated.
* In table 2, what're the differences between the SOTA AD methods and the auto-constructed AD models? Are those SOTA models also have HPS (not necessary network architectures). If not, the comparison is not fair either.

**Documentation:**

N/A

**Limitations:**

As mentioned above, the paper doesn't include the discussion with different applications. The paper can include the discussion with different applications if possible. And the ADGym can be considered as a combination of HPS/NAS work, uncertainty quantification and sensitivity analysis. These points need to be addressed carefully and clearly stated in the work.

**Opportunities For Improvement:**

* As the AD problems have a wide range of application, the paper doesn't include the discussion with a set of different applications. As the needs from different domains, the design choices may be different. For example, the network architectures may be different for different applications. The paper can include the discussion with different applications if possible.
* Another concern of this work is the significance of the Gym work compared with HPS/NAS. As presented in the paper, two of the components, the network construction and network training can be considered as the hyper-parameter search and neural architecture search. And they are also expandable to the data handling stage in case of a standard HPS pipeline.

**Relation To Prior Work:**

Not so clear to me.  Please describe clearly with motivation of the ADGym and its significance in auto-constructing AD models, differentiating from previous HPS/NAS.

**Summary And Contributions:**

The paper proposes a design platform called ADGym for deep anomaly detection (AD) methods. The platform aims to investigate the contribution of each design choice, such as loss functions and network architectures, in detecting anomalies. The paper also explores a wide range of design combinations across multiple large benchmark datasets to evaluate their performance.

---

> ### Author Response · Authors · 2023-08-22
> **Response to Reviewer cdKh (paper revision Aug 22th) -- Part 1**
>
> **Q1**. As the AD problems have a wide range of applications, the paper doesn't include the discussion with a set of different applications. As the needs from different domains, the design choices may be different. For example, the network architectures may be different for different applications. The paper can include the discussion with different applications if possible.
>
> **R1**. We appreciate the thought of exploring the effect of different design choices in various practical applications. Following your suggestion, we group the 29 real-world datasets into 14 domains, including astronautics, biology, finance, et al., and perform a comprehensive evaluation of different design choices on these categorized domains. Detailed results and analysis are presented in Appx. D.3. In general, we find that no design choices are universally superior or inferior, indicating the importance of automated selection of AD design choices. For instance, whereas the GAN-based data augmentation method typically exhibits poorer performance compared to the other methods, it excels in the financial domain. One promising way is to consider domain-specific knowledge, utilizing the semantic meaning of column names in tabular data, which is also one of the future directions we are focusing on.
>
>
> ----
>
> **Q2**. Another concern of this work is the significance of the Gym work compared with HPS/NAS. As presented in the paper, two of the components, the network construction and network training can be considered as the hyper-parameter search and neural architecture search. And they are also expandable to the data handling stage in case of a standard HPS pipeline.
> Please describe clearly the motivation of the ADGym and its significance in auto-constructing AD models, differentiating from previous HPS/NAS.
>
> **R2**. Thanks for your question. ADGym indeed shares the same form of HPS/NAS. We have now enriched this part of related work in Sec. 2.3. But we also want to present the major distinctions: (1) ADGym is not a pure model selection plan for AD. Our original motivation is to design a large benchmark for WSAD to analyze extensive neural-based design choices. Some represent essential components (e.g. loss functions) and under-explored areas (e.g. data augmentation) for AD research.
> (2) Comparing existing HPS/NAS in AD[1], ADGym: (a) presents a more comprehensive selection coverage of design choices beyond network architectures. (b) proposes a more effective paradigm of zero-shot learning for model selection. This means that ADGym uses meta-predictors to predict model (including HPs) construction directly instead of HPS/NAS’s way of selection via evaluation. Our approach is significantly faster to reduce the selection time to just a few seconds.
>
> [1] Li, Yuening, et al. "Autood: Neural architecture search for outlier detection." 2021 IEEE 37th International Conference on Data Engineering (ICDE). IEEE, 2021.
>
> ----
>
> **Q3**. The usage of meta-predictor in both WSAD and SupervisedAD is unclear. Line 162-164 described the meta-predictor as a regression problem, but not well-stated.
>
> **R3**. Thanks for mentioning this.  Following your suggestion, we have now added more details in Section 3.3 and Appx. C.2. to give more clarity to the meta-predictors.
> The meta-predictors we use have 2 sources of input: (1) the learned continuous embedding of the given pipeline (i.e. design choices combination); (2) meta-features of the dataset. The output prediction is the performance score of the pipeline. During inference, the meta-predictors directly predict the rankings of different pipelines without any additional action (e.g. HP tuning). And the meta-predictors can tackle both WSAD and Supervised AD in the same fashion.

---

> > ### Author Response · Authors · 2023-08-22
> > **Response to Reviewer cdKh (paper revision Aug 22th) -- Part 2**
> >
> > **Q4**. In table 2, what're the differences between the SOTA AD methods and the auto-constructed AD models? Are those SOTA models also have HPS (not necessary network architectures). If not, the comparison is not fair either.
> >
> > **R4**. Thanks for your question. The HPs of SOTAs use either default settings in the original paper or are determined by our experience. We specify those settings in Appx. B. Tuning HPs of a fixed model indeed provides performance gains. But this is very hard under weak supervision (not enough labels for hold-out evaluation) and costs a lot of computing. The SS method in Section 4.3 uses supervision for HP (here, referring to all design choices in our paper) selection. It turns out to have inferior performance compared to meta-predictors’ auto-constructed models. ADGym’s auto-constructed models (including their HPs) are specified by the meta-predictor and the whole design space alone, without any tuning, which is more effective and efficient.
> >
> > ----
> >
> > **Q5**. The writing of the paper needs to be improved. The paper needs to be carefully revised to make it more clear and easy to follow. Please describe clearly the motivation of the ADGym and its significance in auto-constructing AD models, differentiating from previous HPS/NAS.
> >
> > **R5**. Thank you for bringing this to our attention. We have refined our paper to enhance clarity and comprehension. To summarize, our research primarily focuses on two objectives: (1) exploring the implications of design choices in neural-network-based AD methods, and (2) developing an automated approach for formulating such design choices.
> > After the revision, section 1 serves as the introduction, section 2 discusses related work, while sections 3.2 and 3.3 delve into the two main objectives. In section 4, which covers experiments, subsections 4.2 and 4.3 present the findings related to our central goals. The paper concludes by addressing limitations and outlining potential future directions.
> >
> > ----
> >
> > We genuinely value your insightful feedback and the time you invested in our paper, elevating its quality and impact.

---

> > > ### Comment · Reviewer_cdKh · 2023-08-26
> > > **Response to Authors**
> > >
> > > While I acknowledge the improvements you have made to the paper, I still have some reservations about the work.
> > > *  I believe that the discussion on the applications of AD problems could be further expanded. Although you have included a comprehensive evaluation of different design choices in various domains, the paper could benefit from a more in-depth exploration of the practical implications of these choices in different scenarios.
> > >
> > > * I still have concerns about the significance of ADGym compared to previous HPS/NAS works. While you have emphasized the differentiating aspects of ADGym, I believe that a more thorough comparison with existing works in the field would help to further establish its uniqueness and value.
> > >
> > > * I found the usage of meta-predictors in both WSAD and SupervisedAD to be unclear, and while you have provided more details in the revised paper, I still have reservations about the effectiveness of this approach.
> > >
> > > * I think that the writing of the paper could be improved further. While you have reorganized the paper to enhance clarity, some sections still feel disjointed, and the motivation and significance of ADGym could be better conveyed throughout the paper.
> > >
> > > To this end, while I acknowledge the efforts you have made to address my concerns, I believe that the paper still has room for improvement in terms of its scope, comparison with existing works, and clarity. Therefore, I will maintain my original score for the paper.

---

> > > > ### Author Response · Authors · 2023-08-28
> > > > **Response to Reviewer cdKh regarding the Latest Comments**
> > > >
> > > > Thank you very much for your thoughtful feedback for ADGym! Your recommendations are helpful for improving the practical significance of ADGym. We have further incorporated the following modifications in the latest revised paper:
> > > >
> > > > ----
> > > >
> > > > 1). We further evaluate the robustness of diverse design choices (which is an important aspect of AD and adversarial ML [1, 2, 3]) under three noisy and corruption settings: i) **Duplicated Anomalies**. In many applications, certain anomalies likely repeat multiple times in the data for reasons such as recording errors [4]; ii) **Irrelevant Features**. Tabular data may contain irrelevant features caused by measurement noise or inconsistent measuring units [5, 6], where these noisy dimensions could hide the characteristics of anomaly data and thus make the detection process more difficult; iii) **Annotation Errors**. In practice,  label noises/contamination caused by annotation errors also present challenges for AD algorithms [7].
> > > >
> > > > We find that Irrelevant Features and Annotation Errors settings significantly impact the performance of various design choices. On the contrary, the Duplicated Anomalies setting improves performance and increases variance, possibly mitigating data imbalance. ResNet architecture consistently performs well, particularly excelling under Duplicated Anomalies. We also conducted experiments with an Irrelevant Features noisy ratio of 0.1 and an Annotation Errors ratio of 0.05, where only a slight mitigation in the impact on design choices, and the conclusions remained consistent with our current findings.
> > > >
> > > > ----
> > > >
> > > > 2). We have also expanded the content [8,9] in the Related Work Section further to explain the difference between HPS/NAS and ADGym. To sum up, the motivation behind ADGym is to (1) propose a comprehensive benchmark that decouples/evaluates various AD design choices from the current SOTA methods, and (2) manages to automatically select/construct AD models by learning the transferable knowledge from the training datasets. The comprehensive benchmark of decoupled AD design choices and exploring other important AD dimensions (e.g., data augmentation and loss function) could be regarded as the major difference between ADGym and HPS/NAS. Again, showing that SOTA methods are not necessarily the panacea and understanding the effect of different design choices, is one of the major contributions of this work; so this work is not an HPS/NAS work, but an automation work built on top of the observations from the benchmark.
> > > >
> > > > ----
> > > >
> > > > 3). Actually, the meta-predictors proposed in our ADGym can be deployed in both weakly- and fully-supervised scenarios. Besides these design choices decoupled from most WSAD methods, we also include those originally designed for the supervised classification tasks such as Binary Cross Entropy (BCE) loss and FTTransformer.
> > > >
> > > > Moreover, we take more effort into decoupling/evaluating the WSAD methods, in order to enhance the overall practicality of ADGym. This is due to the reason that distinguished from the ideal scenario of full supervision, there often exists only a handful of labeled data for model training, while annotating all the input data is often time-consuming and costly[10]. Therefore, ADGym can be applied to a wider range of practical scenarios.
> > > >
> > > > ----
> > > >
> > > > [1] Cai, HanQin, Jialin Liu, and Wotao Yin. "Learned robust pca: A scalable deep unfolding approach for high-dimensional outlier detection." Advances in Neural Information Processing Systems 34 (2021): 16977-16989.
> > > >
> > > > [2] Han, Songqiao, et al. "Adbench: Anomaly detection benchmark." Advances in Neural Information Processing Systems 35 (2022): 32142-32159.
> > > >
> > > > [3] Du, Xuefeng, et al. "Learning diverse-structured networks for adversarial robustness." International Conference on Machine Learning. PMLR, 2021.
> > > >
> > > > [4] Kwon, Donghwoon, et al. "An empirical study on network anomaly detection using convolutional neural networks." 2018 IEEE 38th International Conference on Distributed Computing Systems (ICDCS). IEEE, 2018.
> > > >
> > > > [5] Chang, Chun-Hao, et al. "Data-efficient and interpretable tabular anomaly detection." Proceedings of the 29th ACM SIGKDD Conference on Knowledge Discovery and Data Mining. 2023.
> > > >
> > > > [6] Gopalan, Parikshit, Vatsal Sharan, and Udi Wieder. "Pidforest: anomaly detection via partial identification." Advances in Neural Information Processing Systems 32 (2019).
> > > >
> > > > [7] Nguyen, Duc Tam, et al. "SELF: Learning to Filter Noisy Labels with Self-Ensembling." International Conference on Learning Representations. 2019.
> > > >
> > > > [8] Termritthikun, Chakkrit, et al. "Neural Architecture Search and Multi-Objective Evolutionary Algorithms for Anomaly Detection." 2021 International Conference on Data Mining Workshops (ICDMW). IEEE, 2021.
> > > >
> > > > [9] Kerssies, Tommie. "Neural Architecture Search for Visual Anomaly Segmentation." arXiv preprint arXiv:2304.08975 (2023).
> > > >
> > > > [10] Jiang, Minqi, et al. "Weakly supervised anomaly detection: A survey." arXiv preprint arXiv:2302.04549 (2023).

---

> > > > > ### Comment · Reviewer_cdKh · 2023-08-28
> > > > > **Response to the Authors on Latest Revision**
> > > > >
> > > > > Thank you for your thorough response to my previous concerns. I appreciate the effort you have put into more experiments and revisions.
> > > > >
> > > > > * The inclusion of three noisy and corruption settings (Duplicated Anomalies, Irrelevant Features, and Annotation Errors) demonstrates a thorough understanding of the challenges in anomaly detection and adversarial machine learning.
> > > > > * I am conservative about the points you made on HPS/NAS. Indeed, I don't see the fundamental contributions of ADGym beyond HPS/NAS. The references you mentioned also support the benefits of applying HPS/NAS, not the benchmark under your settings.
> > > > > * I appreciate the flexibility of your meta-predictors, which can be deployed in both weakly- and fully-supervised scenarios. This widens the range of practical scenarios in which ADGym can be applied, making it a more versatile tool for real-world problems.
> > > > >
> > > > > In light of these improvements, I am happy to raise my score for your paper. It's not so innovative in benchmark and dataset, but it has a thorough analysis.

---

### Official Review · Reviewer_ufDM · 2023-07-21
**A fine-grained perspective on design choices for Deep Anomaly Detection**

**Rating:** 6
**Confidence:** 3

**Strengths:**

* By decomposing the AD algorithm into a series of design choices, the authors analyze existing methods from a fine-grained perspective rather than treating the model as a whole. This novel perspective may provide valuable observations that can benefit researchers in this field..
* The paper views model selection as a regression problem and employs a meta-predictor to determine the optimal model from various design choices. This well-designed meta-learning-based method demonstrates remarkable performance improvements over the SOTA methods.
* The authors have released an open-source platform, ADGym, which advances the progress in AD field by providing a convenient and extendable way for researchers to delve into AD methods.

**Additional Feedback:**

In Section 4.3, Table 2 is repeatedly compared with. It would be better to consider a more clear way of presenting the results.

**Clarity:**

This article is well-written overall, but there are several confusing parts:

* In the abstract, it is claimed that "*we may neglect the contribution of other meaningful prerequisite steps like data augmentation and preprocessing by giving all credits to newly designed loss functions and/or architectures*" (lines 8-12). However, in lines 27-28, it is mentioned that *improvements are attributed to novel augmentation functions or loss functions*, and in lines 134-136, it is mentioned that *network architectures and loss functions are often overlooked*. These statements appear contradictory.
* In lines 110-114, it is stated that previous AD model selection frameworks did not fully utilize supervision, while ADGym can leverage these weak signals. However, I cannot see a clear distinction between the proposed method and the method in [53], especially considering that they use the same meta-features. I suggest providing more detailed discussions about their differences.
* The placement of the supplementary material specified in the main text is incorrect.
* The citations in the main text and the supplementary material are inconsistent.
* In Table 3 the process of obtraining results for SS and GT need clarification.
* The experimental settings for Tables 5 and 6 need clarification.

**Correctness:**

* The experiments conducted in Section 4.2 involve a sampling process repeated 1000 times. However, given the vastness of the design space, it remains uncertain whether this quantity of sampling is sufficient to adequately evaluate the performance of the fixed design choice.
* In Section 4.3, the hyperparameters of various baseline methods appear to be fixed. This raises concerns about fairness, as it is natural to adjust hyperparameters to suit different datasets. Additionally, certain design dimensions in ADGym, such as dropout, epochs, and learning rate, are actually hyperparameters but are not fixed in this context.
* Line 264 states that ML meta-predictors are better solutions. However, in Table 5, DL meta-predictors seem to exhibit a significant advantage in a larger design space. The earlier conclusion needs to be adjusted, and an explanation of this phenomenon is required here.

**Documentation:**

The code and data are made publicly available on Github.

**Ethics:**

Not applicable.

**Limitations:**

The author acknowledges a limitation of ADGym, stating that it is currently not suitable for time-series anomaly detection tasks and plans to extend ADGym in the future to address this limitation.

Additionally, it is worth noting that ADGym currently emphasizes anomaly detection under weak supervision (WSAD). Further exploration is needed to investigate the performance under different degrees of supervision, as seen in the author's previous work, ADBench.

**Opportunities For Improvement:**

* The experimental results presented in Sections 4.2 and 4.3 provide the average performance of various methods across 29 datasets. Given the substantial variations in numerical values across these datasets, the relative rankings of the methods appear to be a more reliable indicator. More importantly, this paper lacks a comprehensive discussion on the correlation between the effectiveness of different design choices and the characteristics of the datasets. The majority of experiments conducted in this paper show no significant differences among the design choices, which is expected considering the overall performance across 29 datasets. As a result, the practical guidance offered by these findings is limited. It would be more beneficial to provide insights into the preferences of different dataset characteristics for specific design choices.
* The meta-predictor consistently outperforms the SOTA models but falls short compared to GT. In this context, is the GT referring to the best model among the 1000 samples or the best model in the entire design space? (I prefer the former). To further evaluate the capabilities of the meta-predictor, an analysis of the trade-off between efficiency and performance can be conducted, shedding light on its strengths and limitations.

**Relation To Prior Work:**

This article clearly explains the relation to previous works in Section 2.

**Summary And Contributions:**

This paper presents ADGym, an open-source platform designed for evaluating and automatically selecting finer-grained design choices. By breaking down existing methods into multiple design dimensions, the paper offers a comprehensive assessment of each design choice. Additionally, the paper proposes a meta-learning-based automatic model selection approach to construct appropriate models for new datasets, consistently surpassing state-of-the-art (SOTA) models. Through a meticulous analysis of existing methods from a fine-grained perspective, this paper provides valuable insights into the field.

---

> ### Author Response · Authors · 2023-08-22
> **Response to Reviewer ufDM (paper revision Aug 22th) -- Part 1**
>
> **Q1**. The experimental results presented in Sections 4.2 and 4.3 provide the average performance of various methods across 29 datasets. Given the substantial variations in numerical values across these datasets, the relative rankings of the methods appear to be a more reliable indicator.
>
> **R1**. Thanks for your kind reminder. Now we have provided both numerical values and their relative rankings of AUCROC and AUCPR metrics for better evaluating different design choices and meta-predictors, as demonstrated in Appx. D.1.
>
> ----
>
> **Q2**. More importantly, this paper lacks a comprehensive discussion on the correlation between the effectiveness of different design choices and the characteristics of the datasets. The majority of experiments conducted in this paper show no significant differences among the design choices, which is expected considering the overall performance across 29 datasets. As a result, the practical guidance offered by these findings is limited. It would be more beneficial to provide insights into the preferences of different dataset characteristics for specific design choices.
>
> **R2**. Great point! As you suggested, we further conduct extensive experiments to analyze this correlation from the following two aspects (results in Appx. D.2 and D.3):
>
> 1. **Detection performance of different design choices on various types of anomalies**.
> For AD, anomalous patterns are typically manifested in the types of anomalies present in the dataset. Following our previous work ADBench[1], we generate four types of synthetic anomalies, including local, global, dependency, and clustered anomalies, for further evaluating different design choices. We find that (1) In most cases, data augmentation methods still have negative impacts on detection performance, whereas GAN-based methods may hold the potential for improving model performance on specific anomalies, e.g., it significantly improves both AUCROC and AUCPR on dependency anomalies w.r.t. la=10. (2) ResNet architecture shows superiority on the dependency anomalies, while the AutoEncoder is also competitive on both local and global anomalies; (3) The deviation loss consistently outperforms other counterparts across different types of anomalies. Moreover, we verified that the meta-predictors are capable of learning these important findings. Details are shown in Appx. D.2.
>
> 2. **Detection performance of different design choices on application domains of the datasets**.
> We group the 29 real-world datasets into 14 domains, such as astronautics, biology, and finance. No design choices consistently dominate (or lose) across all domains, indicating the importance of automated selection of design choices. For instance, despite GAN-based data augmentation method underperforming in most domains, it excels in the financial domain. Details are shown in Appx. D.3.
>
> [1] Han, Songqiao, et al. "Adbench: Anomaly detection benchmark." Advances in Neural Information Processing Systems 35 (2022): 32142-32159.
>
> ----
>
> **Q3**. The meta-predictor consistently outperforms the SOTA models but falls short compared to GT. In this context, is the GT referring to the best model among the 1000 samples or the best model in the entire design space? (I prefer the former).
>
> **R3**. Thanks for the correction. The GT actually refers to the best performance among the 1000 randomly selected design choices. We have provided more details about the baselines of proposed meta-predictors to avoid confusion, as illustrated in Section 4.3.
>
> ----
>
> **Q4**. To further evaluate the capabilities of the meta-predictor, an analysis of the trade-off between efficiency and performance can be conducted, shedding light on its strengths and limitations.
>
> **R4**. We appreciate the valuable advice for exploring the trade-off between the efficiency and performance of meta-predictors. We treat this as one of the long-term goals of ADGym, and plan to integrate multi-task learning (MTL)[1,2,3] into the model training of meta-predictor, e.g., teaching the meta-predictor to learn how to balance between choosing better (and usually deeper) network architectures like Transformer or faster architectures like MLP based on the characteristics of the input dataset. Our current meta-predictor can quickly extract meta-features (less than 1 minute for the majority of the datasets) and predict the best K designs. This greatly reduces the time and cost users spend on designing AD pipelines and tuning model hyperparameters.
>
> [1] Kumar, Ananya, et al. "Fine-tuning can distort pretrained features and underperform out-of-distribution." arXiv preprint arXiv:2202.10054 (2022).
>
> [2] Yin, Mingzhang, et al. "Meta-Learning without Memorization." International Conference on Learning Representations. 2019.
>
> [3]Vicol, Paul, Luke Metz, and Jascha Sohl-Dickstein. "Unbiased gradient estimation in unrolled computation graphs with persistent evolution strategies." International Conference on Machine Learning. PMLR, 2021.

---

> > ### Author Response · Authors · 2023-08-22
> > **Response to Reviewer ufDM (paper revision Aug 22th) -- Part 2**
> >
> > **Q5**. The author acknowledges a limitation of ADGym, stating that it is currently not suitable for time-series anomaly detection tasks and plans to extend ADGym in the future to address this limitation.
> > Additionally, it is worth noting that ADGym currently emphasizes anomaly detection under weak supervision (WSAD). Further exploration is needed to investigate the performance under different degrees of supervision, as seen in the author's previous work, ADBench.
> >
> > **R5**. Great point! For most AD scenarios, both i) the ratio of labeled anomalies (rla for short, used in ADBench[1] and previous study DeepSAD[2]. For example, rla=10% means that 10% anomalies in the train set are known while other samples remain unlabeled) and ii) the number of labeled anomalies (nla for short, used in ADGym, previous studies REPEN [3] and DevNet [4]) can be regarded as good measurements of varying degrees of supervision, while we consider the latter is more suitable for our proposed ADGym.
> >
> > The main reason is that for the first case if one wants to obtain the optimal design choices for a newcoming dataset via the trained meta-predictor, he has to first collect [meta features, rla, embedding of AD components] as the input vector of meta-predictor. However, defining rla as the ratio of labeled anomalies to the total number of anomalies presents a challenge, as the latter is often unknown in real-world scenarios. Therefore, nla exhibits more transferability in the task of automatic model selection, as it expresses identical supervision across different datasets. In future work, we plan to discuss ADGym in both of the above situations, therefore enhancing the generality of ADGym in the WSAD problem.
> >
> > [1] Han, Songqiao, et al. "Adbench: Anomaly detection benchmark." Advances in Neural Information Processing Systems 35 (2022): 32142-32159.
> >
> > [2] Ruff, Lukas, et al. "Deep Semi-Supervised Anomaly Detection." International Conference on Learning Representations. 2019.
> >
> > [3] Pang, Guansong, et al. "Learning representations of ultrahigh-dimensional data for random distance-based outlier detection." Proceedings of the 24th ACM SIGKDD international conference on knowledge discovery & data mining. 2018.
> >
> > [4] Pang, Guansong, Chunhua Shen, and Anton van den Hengel. "Deep anomaly detection with deviation networks." Proceedings of the 25th ACM SIGKDD international conference on knowledge discovery & data mining. 2019.
> >
> > ----
> >
> > **Q6**. The experiments conducted in Section 4.2 involve a sampling process repeated 1000 times. However, given the vastness of the design space, it remains uncertain whether this quantity of sampling is sufficient to adequately evaluate the performance of the fixed design choice.
> >
> > **R6**. Thank you for mentioning this. Indeed, the computational resources required for ADGym are quite substantial. For example, even if the sampling process of different design choices is conducted only 1,000 times, this would result in a total number of 29,000 models to be trained on 29 real-world datasets, which requires training on multiple GPU clusters for several days, not to mention the inclusion of some complex deep learning models like FTTransformer and GAN in the design space.
> >
> > Despite that, we conducted three independent experiments under different random seeds (resulting in 87,000 times model training). We observe that similar conclusions can be drawn for each independent experiment, indicating this quantity is fairly enough. For our long-term plan, we plan to sample more design choices for learning more effective meta-predictors, since our experimental results in Section 4.3 indicate that larger design space is often helpful for constructing more powerful AD models.
> >
> > ----
> >
> > **Q7**. In Section 4.3, the hyperparameters of various baseline methods appear to be fixed. This raises concerns about fairness, as it is natural to adjust hyperparameters to suit different datasets. Additionally, certain design dimensions in ADGym, such as dropout, epochs, and learning rate, are actually hyperparameters but are not fixed in this context.
> >
> > **R7**. Thanks for mentioning this. We want to clarify that the hyperparameters (i.e., design choices) of AD models are automatically selected via the proposed meta-predictors, which are learned based on the knowledge of historical training datasets. Instead of tuning on the validation set, we manage to teach the meta-predictor how to select the optimal design choices, therefore it can perform zero-shot learning on the newcoming dataset.
> >
> > Besides, we consider using the default hyperparameters of baseline AD methods, as these hyperparameter settings often correspond to the best-performing experimental results in the original papers, while tuning on a small number of labeled data often leads to relatively inferior performance (e.g., the SS baseline method illustrated in Section 4.3 ).

---

> > > ### Author Response · Authors · 2023-08-22
> > > **Response to Reviewer ufDM (paper revision Aug 22th) -- Part 3**
> > >
> > > **Q8**. Line 264 states that ML meta-predictors are better solutions. However, in Table 5, DL meta-predictors seem to exhibit a significant advantage in a larger design space. The earlier conclusion needs to be adjusted, and an explanation of this phenomenon is required here.
> > >
> > > **R8**. Thanks so much for pointing it out! We have now revised Section 4.3 (especially question 5) to clarify this. We find that ML meta-predictors perform better in relatively smaller design space while DL meta-predictors achieve better results for larger design space, and both of them benefits from larger design space where the AUCROC and AUCPR are improved. This may be due to the reason that larger design space is more helpful for DL meta-predictors to prevent overfitting on historical datasets. However, the performance differences between these two types of meta-predictors are actually very small (say 0.003~0.008). Considering ML meta-predictors often converge faster and require fewer computation resources, we recommend them as the final solutions.
> > >
> > > ----
> > >
> > > **Q9**. In the abstract, it is claimed that "we may neglect the contribution of other meaningful prerequisite steps like data augmentation and preprocessing by giving all credits to newly designed loss functions and/or architectures" (lines 8-12). However, in lines 27-28, it is mentioned that improvements are attributed to novel augmentation functions or loss functions, and in lines 134-136, it is mentioned that network architectures and loss functions are often overlooked. These statements appear contradictory.
> > >
> > > **R9**. We appreciate your clarification. We have now revised those arguments to address the inconsistency. Originally, we just wanted to show that different researchers have very different beliefs about the weight of different design dimensions in developing SOTA AD methods.
> > >
> > > ----
> > >
> > > **Q10**. In lines 110-114, it is stated that previous AD model selection frameworks did not fully utilize supervision, while ADGym can leverage these weak signals. However, I cannot see a clear distinction between the proposed method and the method in [53], especially considering that they use the same meta-features. I suggest providing more detailed discussions about their differences.
> > >
> > > **R10**. Thank you so much for these suggestions. We have now enriched this part of related work in Section 2.3.1 and provide more details of meta-predictors in Section 3.3 and Appx. C.2.  ADGym’s meta-predictors indeed share the same meta-learning idea with MetaOD[1]. As we are the same group of authors with the same vision of automatic and reproducible AD systems. But we also want to present some major distinctions:
> > >
> > > (1) ADGym’s meta-predictors have 3 sources of input: (a) weak supervision (i.e. number of anomalies in the given dataset); (b) the embedding of the given pipeline (i.e. design choices combination); (c) meta-features of the dataset. Only the last input shares the same form with MetaOD. ADGym also provides another framework of meta-learning, the end-to-end meta-predictors. This kind of meta-predictors will not explicitly define meta-features.
> > >
> > > (2) MetaOD focuses on selecting classical unsupervised methods. ADGym focuses on weakly-supervised neural models and has a much greater selection space to take into account.
> > >
> > > [1] Zhao, Yue, Ryan Rossi, and Leman Akoglu. "Automatic unsupervised outlier model selection." Advances in Neural Information Processing Systems 34 (2021): 4489-4502.

---

> > > > ### Author Response · Authors · 2023-08-22
> > > > **Response to Reviewer ufDM (paper revision Aug 22th) -- Part 4**
> > > >
> > > > **Q11**. The placement of the supplementary material specified in the main text is incorrect.
> > > >
> > > > **R11**. Thanks for this question. We examined the paper carefully and addressed the Appx.  reference issues, e.g., (1) In Section 3.3, “We defer more details to Appx. C.” is revised to “We defer more details to Appx. C.2.”. (2) In Section 4.1, “We defer the details of datasets and baselines in Appx. B.” is revised to “We defer the details of datasets and baselines in Appx. A and B.”
> > > > If there are still similar issues that we missed, we will be grateful to receive additional feedback.
> > > >
> > > > ----
> > > >
> > > > **Q12**. The citations in the main text and the supplementary material are inconsistent.
> > > >
> > > > **R12**.  Thanks for pointing out this. We examined the paper carefully and addressed the following citation issues: (1) two inconsistent citations of ResNet are corrected into one. (2)
> > > > If there are still similar issues that we missed, we will be grateful to receive additional feedback.
> > > >
> > > > ----
> > > >
> > > > **Q13**. In Table 3 the process of obtaining results for SS and GT need clarification.
> > > >
> > > > **R13**. Thanks for pointing out this. We have now explained this in the first paragraph of Section 4.3.
> > > >
> > > > ----
> > > >
> > > > **Q14**. The experimental settings for Tables 5 and 6 need clarification.
> > > >
> > > > **R14**. Thanks for mentioning this. Now we have added more details of the experimental settings for Table 5 and 6, as illustrated in question 5 and 6 in Section 4.3, respectively.
> > > >
> > > > ----
> > > >
> > > > **Q15**. In Section 4.3, Table 2 is repeatedly compared with. It would be better to consider a more clear way of presenting the results.
> > > >
> > > > **R15**. We want to thank you again for these detailed comments. We have now revised Section 4.3 based on your suggestions.
> > > >
> > > > ----
> > > >
> > > > ADGym has been significantly improved by these great thoughts, and we truly appreciate your devotion to making ADGym better! Hats off!

---

> > > > > ### Comment · Reviewer_ufDM · 2023-08-25
> > > > >
> > > > > Thank you for providing the detailed response. The response has addressed most of my concerns. I will slightly increase my rating.
> > > > > The results in Appx. D.2 and D.3 are valuable additions, and conducting more detailed and comprehensive experiments would be beneficial for the field in the future. Additionally, I couldn't find the mentioned revisions in Q9.

---

> > > > > > ### Author Response · Authors · 2023-08-26
> > > > > > **Response to Reviewer ufDM for addressing Q9**
> > > > > >
> > > > > > Thank you for your thoughtful feedback and for recognizing the value of our additional results. We're pleased to hear that our revisions have largely addressed your concerns.
> > > > > >
> > > > > > Regarding Q9, we appreciate the chance to clarify. We have aimed to showcase the diversity involved in designing anomaly detection methods, which includes not only commonly discussed design dimensions in the previous studies, like network architectures and loss functions, but also those that are often overlooked, like data augmentation and preprocessing. Corresponding to the above clarification, we have made the following modifications to the paper:
> > > > > >   - We have revised the description on lines 26-27, changing the original 'novel augmentation functions' to 'novel augmentation functions or loss functions'.
> > > > > >   - We have removed the confusing expression on line 142.
> > > > > >
> > > > > > Thank you very much for your suggestions;  we hope that the revised version will eliminate any confusion for the readers.

---

### Official Review · Reviewer_n2uJ · 2023-07-21
**A novel benchmark but lack of sufficient details.**

**Rating:** 7
**Confidence:** 5
**Correctness:** Yes, I find the claims made in the pa…
**Clarity:** Yes, the paper is well-structured and…

**Strengths:**

The paper verifies the effect of different component choices in the AD algorithm on performance through a large number of experiments. The experiments are conducted on a wide range of datasets. The observations concluded in the paper is instructive for future work on designing effective AD algorithm.
2. The proposed framework, ADGym, is verified on 29 datasets, and the experimental results show that the automatically constructed models can surpass SOTA methods in various settings.
3. An open-source platform is built based on the proposed ADGym, which is valuable for the AD community and helpful for further advancements in automatic AD model generation.

**Additional Feedback:**

See Above.

**Documentation:**

The source code is provided.
The provided documentation meets the requirements of the D&B track.

**Ethics:**

 No, there are no or only very minor ethics concerns

**Limitations:**

The large evaluations on AD design choices brings a lot of useful findings, like MLP is more competitive, BCE loss is better for network training, etc.
However, it seems that the proposed ADGym does not make use of the previous findings, though ADGym achieves competitive performance.

**Opportunities For Improvement:**

To improve readability and provide more insightful analysis, I suggest the following revision:
1. Provide a suitable introduction for meta-feature and embedding of AD component in Sec. 3.3
2. Provide the detailed experimental settings of the figures, like Figure 2,3,4
3. In Lines 237-240, it would be better to provide the location of the experimental results of ADGym, maybe Table 3.
4. In Lines 212-214, the authors attribute the unsatisfactory performances of ResNet and Transformer to the mismatch between the complicated model and the limited labeled data. What does the word “complicated” refer to? Complicated architecture or the large number of parameters? More discussion of the failures of ResNet and Transformer should be provided.

**Relation To Prior Work:**

Yes, the authors provide a detailed discussion of related works.

**Summary And Contributions:**

This paper focuses on anomaly detection and provides an understanding of two problems:
1) Which components of an AD algorithm matter?
2)How to build suitable algorithms for new AD tasks automatically?

Existing surveys or benchmarks for anomaly detection only focus on different AD algorithms but ignore the effectiveness of fine-grained components.
This paper breaks down AD algorithms into fine-grained components and provides a comprehensive understanding of the impact of different design choices on the final performance.
Based on the evaluations, the authors present helpful findings, 1) for network architectures, MLP is more competitive than ResNet and FTTransformer, 2) BCE loss and Adam optimizer are better for network training, etc.
Besides, the paper proposes a new framework, ADGym, to automatically select optimal design choices for unseen AD tasks.
The experimental results show that the strategies constructed by ADGym achieve promising performance.

---

> ### Author Response · Authors · 2023-08-22
> **Response to Reviewer n2uj (paper revision Aug 22th)**
>
> **Q1**. To improve readability and provide more insightful analysis, I suggest the following revision:
> 1. Provide a suitable introduction for meta-feature and embedding of AD component in Sec. 3.3.
> 2.  Provide the detailed experimental settings of the figures, like Figure 2,3,4.
> 3.  In Lines 237-240, it would be better to provide the location of the experimental results of ADGym, maybe Table 3.
> 4.  In Lines 212-214, the authors attribute the unsatisfactory performances of ResNet and Transformer to the mismatch between the complicated model and the limited labeled data. What does the word “complicated” refer to? Complicated architecture or the large number of parameters? More discussion of the failures of ResNet and Transformer should be provided.
>
> **R1**. Thanks for your sincere and thorough suggestions! We have now revised the corresponding sections.
>
> (1) Sec. 3.3 is revised to present the meta-learning process more clearly, where the meta-feature and component embedding is generally introduced. More details are added in Appx. C.2.
>
> (2) We provide more detailed experimental settings in Sec. 4.2.1, 4.2.2, and 4.2.3.
>
> (3) We add the sentence “The central experimental results of ADGym’s performance is provided in Table 3” in the beginning paragraph of Sec. 4.3.
>
> (4) More discussion on the evaluation of ResNet and Transformer is provided in Sec. 4.2.2. Here, we originally meant that ResNet and Transformer are more complex in architecture design. We have now clarified this in Sec. 4.2.2.
>
> Thanks again! These revisions will make ADGym a more accessible and complete work.
>
> ----
>
> **Q2**. The large evaluations on AD design choices brings a lot of useful findings, like MLP is more competitive, BCE loss is better for network training, etc. However, it seems that the proposed ADGym does not make use of the previous findings, though ADGym achieves competitive performance.
>
> **R2**. Thanks for your valuable question! We report those interesting findings according to the average performance of those design choices. In practice, we hope solutions generated by ADGym to be robust for any incoming dataset and application. So we did not insert specific inductive bias for ADGym by utilizing those findings. We have also provided a slight discussion on this in Sec. 4.3, question 6.
> It is definitely a fantastic idea to capitalize on these findings. Firstly, we may add an empirical guidance section to share them directly in the future. Secondly, we will try to incorporate those feedback into ADGym’s meta-predictor. We will work on this perspective to make ADGym a more flexible and practical system.

---

> > ### Comment · Reviewer_n2uJ · 2023-08-24
> >
> > I thank the authors for addressing my concerns. The responses have addressed my concerns. Thus, I decide to keep my initial high score.

---

### Official Review · Reviewer_YA3B · 2023-07-23
**A new toolbox for anomaly detection on tabular data with interesting findings in experiments.**

**Rating:** 6
**Confidence:** 5
**Clarity:** The paper is well-structured, clear, …

**Strengths:**

1. This work effectively breaks down AD algorithms into separate components, conducting comprehensive experiments to evaluate each one. The finding that data augmentation contributes little to performance enhancement is intriguing.

2. Compared to existing AD toolkits, such as PyOD and PyODDS, ADGym offers greater flexibility by allowing for the selection and combination of different components, ensuring fairer comparisons between models.

3. ADGym introduces the first automatic model-building framework for weak supervision, a scenario where only very few labeled samples are available for model training.

**Additional Feedback:**

1. Could the authors clarify why certain design dimensions in Table 1 are highlighted (in bold), while others are not?

2. What are the design choices for machine learning-based models, such as XGBoost, as shown in Table 3?

3. What is the detailed architecture of the Autoencoder?

**Correctness:**

The evaluation methods and experiment design appropriate and performed correctly.

**Documentation:**

The associated GitHub repository is well-documented and user-friendly.

**Ethics:**

There do not appear to be any major ethical concerns associated with this work.

**Limitations:**

The paper does not explicitly address any limitations. From my viewpoint, a major limitation is that the experiments only focus on the weakly-supervised scenario.

**Opportunities For Improvement:**

1. This paper concentrates mainly on deep AD model design choices, seemingly overlooking the conventional machine learning models. While the focus on advanced models is understandable, the absence of traditional models such as tree ensembles - which have proven to be competitive, as demonstrated in ADBench and displayed in Table 3 - limits the comprehensiveness of the study.


2. The design spaces in Table 1 present room for improvement. For instance, the design of the hidden layers appears somewhat arbitrary, and the relationship between the number of layers and the hidden dimensions isn't explicitly clear. Could these parameters be treated independently for optimization?  Another point of ambiguity involves potential conflicts between different configurations.  For instance, the question remains as to whether certain network architectures may require a specific loss function, thus introducing a constraint in design choices.

**Relation To Prior Work:**

It clearly discussed the difference between previous AD benchmarks and toolkits.

**Summary And Contributions:**

This paper presents ADGym, a new benchmark and toolbox for anomaly detection (AD) algorithm evaluation and selection. Its unique feature is that it separates the different aspects of deep AD algorithms, such as loss functions and network architectures, allowing for a more precise examination of each component. Additionally, ADGym can assess the effect of data augmentation and preprocessing. Experimental results demonstrate that AD methods built using ADGym significantly outperform current state-of-the-art methods, particularly in weakly supervised scenarios.

---

> ### Author Response · Authors · 2023-08-22
> **Response to Reviewer YA3B (paper revision Aug 22th) -  Part 1**
>
> **Q1**. This paper concentrates mainly on deep AD model design choices, seemingly overlooking the conventional machine learning models. While the focus on advanced models is understandable, the absence of traditional models such as tree ensembles - which have proven to be competitive, as demonstrated in ADBench and displayed in Table 3 - limits the comprehensiveness of the study.
>
> **R1**. You are absolutely right. Traditional ML methods like tree-based models are a strong baseline in any practical scenario. The current ADGym primarily centers around neural-network-based methods. We will try to analyze and decompose ML-based models, especially tree-based models in the future (see Section 5) to expose a more comprehensive version of ADGym.
>
> ----
>
> **Q2**.  The design spaces in Table 1 present room for improvement. For instance, the design of the hidden layers appears somewhat arbitrary, and the relationship between the number of layers and the hidden dimensions isn't explicitly clear. Could these parameters be treated independently for optimization?
> Another point of ambiguity involves potential conflicts between different configurations. For instance, the question remains as to whether certain network architectures may require a specific loss function, thus introducing a constraint in design choices.
>
> **R2**. Thanks for mentioning this! In ADGym, the primary goal is to explore more important/interesting design dimensions under limited computational costs since we have to train and evaluate thousands of combined AD methods on nearly 30 real-world datasets. Therefore, we place more emphasis on the design dimensions like data augmentation methods and network architectures, constraining the design space of other dimensions like hidden layers and hidden dimensions (theoretically speaking, these design spaces can be expanded infinitely). In future work, we plan to expand the design space, conduct a more extensive sampling of design choices, and utilize more computational resources to build more powerful meta-predictors.
>
> In addition, we want to clarify that ADGym actually considers the potential conflicts between different configurations. For example, the inverse loss function and batch resampling method are regarded as conflicting (hence are removed from the entire design space), since the inverse loss is solely designed for extremely class-imbalance scenarios. We also impose restrictions on the activation functions (e.g., ReLU for FTTransformer) of DL-based models, since inappropriate activation functions may lead to non-convergence of model training.
>
> ----
>
> **Q3**. The paper does not explicitly address any limitations. From my viewpoint, a major limitation is that the experiments only focus on the weakly-supervised scenario.
>
> **R3**. Following your concern, we have rewritten Section 5 to express the limitations of our current work explicitly.
> From our point of view, traditional unsupervised AD methods have already been thoroughly benchmarked in our previous work ADBench [1]. The corresponding model selection methods are also developed [2]. In comparison, the utilization of weak supervision is still a point to be explored, and their performance has often been SOTA. So we focus on weakly-supervised scenarios in this work.
> Additionally, unlike DL-based methods, the majority of traditional unsupervised AD models do not share universal design principles (e.g. use of neural networks). This results in a vast challenge to decompose them with consistency and completeness. As a long-term open-source project, we will try to extend the scope and incorporate more detection methods (and categories) in the future.
>
> [1] Han, Songqiao, et al. "Adbench: Anomaly detection benchmark." Advances in Neural Information Processing Systems 35 (2022): 32142-32159.
>
> [1] Zhao, Yue, Ryan Rossi, and Leman Akoglu. "Automatic unsupervised outlier model selection." Advances in Neural Information Processing Systems 34 (2021): 4489-4502.
>
> ----
>
> **Q4**. Could the authors clarify why certain design dimensions in Table 1 are highlighted (in bold), while others are not?
>
> **R4**. Thanks for mentioning this. Bolded design dimensions are those of greater impact on the AD task. This is discussed in Section 4.2. We have now updated the caption of Table 1 and revised the introduction to explain this.

---

> > ### Author Response · Authors · 2023-08-22
> > **Response to Reviewer YA3B (paper revision Aug 22th) - Part 2**
> >
> > **Q5**. What are the design choices for machine learning-based models, such as XGBoost, as shown in Table 3?
> >
> > **R5**. Thanks for the question. We would clarify this confusion. The reported performance in Table 3 is the evaluation of different model selection plans, especially different meta-predictors. The mention of the tree-based model here is for the choice of the meta-predictor but not the anomaly detection itself. Note the meta-predictor makes predictions on the performance rank of different design choices. We apologize for the confusion caused by our lack of clarity in the exposition.
> >
> > ----
> >
> > **Q6**. What is the detailed architecture of the Autoencoder?
> >
> > **R6**. Thanks for mentioning this. We adopt the auto-encoder architecture in [1], with hyperparameters consistent with the ADGym. We now specify this design choice in Appx.  C.1.
> >
> > [1] Zhou, Yingjie, et al. "Feature encoding with autoencoders for weakly supervised anomaly detection." IEEE Transactions on Neural Networks and Learning Systems 33.6 (2021): 2454-2465.

---

> > > ### Comment · Reviewer_YA3B · 2023-08-27
> > > **Thank the authors for their response.**
> > >
> > > I have reviewed the rebuttals and other feedback. While the system design of ADGym appears to be preliminary (for instance, a limited range of base models have been considered, and the hyperparameter search space seems arbitrary), I believe that **ADGym represents a significant exploration in a very practical direction**. The quality of the code is commendable. On balance, I've decided to maintain my original score and encourage the authors to further refine ADGym.

---

### Official Review · Reviewer_hpUe · 2023-07-28
**A design choice benchmark for Weakly Supervised Anomaly Detection.**

**Rating:** 6
**Confidence:** 4
**Correctness:** Yes.
**Clarity:** Yes.

**Strengths:**

- The proposition shifts the perspective from the model to the pipeline which imho actually reflects the real-world scenarios.
- Extensive experiments are done to show the effectiveness of the meta predictor.
- The axes of design choices is reasonably large.

**Additional Feedback:**

See "Opportunities For Improvement"

**Documentation:**

This work doesn't propose a new dataset.

**Limitations:**

See "Opportunities For Improvement"

**Opportunities For Improvement:**

- I think one axis that is left is having unsupervised pre-training. IMHO this is very important, especially for the DL-based methods. It would be great if the authors could comment on this.
- There is heavy use of meta-predictor in the paper but very little discussion in the related work section.
- Finally, a lot of design choices made for the benchmark are not explained. For instance, authors say in lines 174-175 "Datasets with sample sizes smaller than 1000, as well as those with problematic model results, are removed, resulting in a total of 29 remaining datasets, as is shown in Appx. Table 1" but there is no explanation as to why was it done especially the smaller size datasets? Isn't that the real-world setting?

**Relation To Prior Work:**

I think the authors should include a section on meta-predictors in the related work section as this is one of the main contributions of this work but very little to no discussion is present in the related work section.

**Summary And Contributions:**

The authors propose ADGym, a design platform for large evaluation and automatic selection of Anomaly Detection (AD) design choices. The authors propose to look at the model pipeline ( data handling → network construction → network training) instead of models in tandem for Weakly Supervised Anomaly Detection (WSAD). The authors also propose a meta-predictor that predicts the performance of a design choice configuration on a given dataset. Extensive experiments are done to validate the claim.

---

> ### Author Response · Authors · 2023-08-22
> **Response to Reviewer hpUe (paper revision Aug 22th)**
>
> **Q1**. I think one axis that is left is having unsupervised pre-training. IMHO this is very important, especially for the DL-based methods. It would be great if the authors could comment on this.
>
> **R1**. Thanks for sharing this thought. Following your advice, we explore the impact of unsupervised pre-training on the model performance of downstream WSAD tasks, as shown in Appx.  D.1.3 (codes are also updated in ADGym GitHub repository). Following [1, 2], we construct AutoEncoder corresponding to different network architectures like ResNet and FTTransformer, and employ unsupervised reconstruction loss to perform model pre-training. Unlike NLP and CV tasks that contain rich textual semantics and visual patterns [3], we do not observe a significant advantage of pre-training over other network initialization methods like Xavier and Kaiming, probably due to the reason that the inherent structure of tabular data is more rigid and constrained, resulting in the model struggling to learn general features without label guidance.
>
> [1] Ruff, Lukas, et al. "Deep one-class classification." International conference on machine learning. PMLR, 2018.
>
> [2] Ruff, Lukas, et al. "Deep Semi-Supervised Anomaly Detection." International Conference on Learning Representations. 2019.
>
> [3] Han, Songqiao, et al. "Adbench: Anomaly detection benchmark." Advances in Neural Information Processing Systems 35 (2022): 32142-32159.
>
> ----
>
> **Q2**. There is heavy use of meta-predictor in the paper but very little discussion in the related work section.
> I think the authors should include a section on meta-predictors in the related work section as this is one of the main contributions of this work but very little to no discussion is present in the related work section.
>
> **R2**. Thanks for mentioning this. Following your suggestion, we enrich the literature on meta-learning for AD model selection in Section 2.3.1.
>
> ----
>
> **Q3**. Finally, a lot of design choices made for the benchmark are not explained. For instance, authors say in lines 174-175 "Datasets with sample sizes smaller than 1000, as well as those with problematic model results, are removed, resulting in a total of 29 remaining datasets, as is shown in Appx. Table 1" but there is no explanation as to why it was done, especially the smaller size datasets? Isn't that the real-world setting?
>
> **R3**. Thanks for your question. We agree with your opinion that in real-world AD scenarios, one also needs to develop AD models on small-scale datasets (say, fewer than 1,000). In such cases, traditional ML-based AD models/automatic model selection methods could be served as effective solutions, as explained in our previous works MetaOD [1] and ADBench [2].
>
> We want to clarify that the current ADGym mainly focuses on the evaluation and automatic selection of different design choices involved in designing DL-based AD methods. Most of them (e.g. GAN-based methods in data augmentation and FTTransformer architecture) require an adequate amount of input data for model training. Therefore, we removed those datasets smaller than 1,000 to ensure that the model parameters converged in most cases. We also removed those datasets that cause errors (e.g. ResNet often fails in the cardio dataset) when performing model training and evaluation.
>
> [1] Zhao, Yue, Ryan Rossi, and Leman Akoglu. "Automatic unsupervised outlier model selection." Advances in Neural Information Processing Systems 34 (2021): 4489-4502.
>
> [2] Han, Songqiao, et al. "Adbench: Anomaly detection benchmark." Advances in Neural Information Processing Systems 35 (2022): 32142-32159.

---

> > ### Author Response · Authors · 2023-08-28
> > **Response to Reviewer hpUe before the end of discussion**
> >
> > Dear Reviewer hpUe,
> >
> > Since the End of author/reviewer discussions is just in one day, may we know if our response addresses your main concerns? If so, we kindly ask for your reconsideration of the score.
> >
> > Should you have any further advice on the paper and/or our rebuttal, please let us know and we will be more than happy to engage in more discussion and paper improvements. We would really appreciate it if our next round of communication could leave time for us to resolve any of your remaining or new questions.
> >
> > Thank you so much for devoting time to improving our benchmark!

---

### Author Response · Authors · 2023-08-22
**Summary of Our Responses and Long-term Plan**

We sincerely thank all the reviewers for their encouraging and insightful comments. We have carefully read through them and provided corresponding responses individually.

We upload the revised paper and the supplementary material, with changes highlighted in blue. We provide the single pdf with the main content + supplementary material in the supplementary.

The primary changes are summarized below:

- **Related Work**: We add a new section on the use of meta-predictors and HPS/NAS in the related work section (from Reviewer hpUe and cdKh).

- **More Evaluation**: (1) Per Reviewer ufDM's feedback, we have added results using the relative ranks of AUCROC and AUCPR in Appx. D.1.2 and D.4.2. (2) Addressing comments from Reviewers ufDM and cdKh, we expanded our analysis across different dataset domains in Appx.  D.3. (3) Based on Reviewer ufDM's suggestion, we examined over 300 synthetic datasets with distinct anomaly types in Appx. D.2.1. (4) We add experiments of ADGym on the global anomaly synthetic dataset in Appx. D.2.2, as advised by Reviewer ufDM. (5) Acting on Reviewer hpUe's input, we have included results using an unsupervised pre-trained model on datasets with 10 anomalies in Appx. D.1.3.(6) Based on Reviewer cdKh's suggestion, we have added evaluations for design choices under three noisy and corruption settings (Duplicated Anomalies, Irrelevant Features, and Annotation Errors) in Appx. D.5.

- **Paper Writing**: (1) We provide more details on the meta-predictors of ADGym in Section 3.3 and Appx. C.2, e.g., how to utilize supervision (from Reviewer cdKh and n2uj). (2) We revise the Introduction to distinguish our work from previous HPS/NAS (from Reviewer cdKh). (3)We revise Section 4.1 to make the experimental settings more clear (from Reviewer hpUe and ufDM). (4) We revise and state the limitations of ADGym more clearly in Section 5 (from Reviewer YA3B). (5) We introduce the design dimensions highlighted in Table 1 in line 31 (from Reviewer YA3B). (6) We revise Section 2.3 to enrich the literature on meta-learning for AD model selection.(from Reviewer cdKh and hpUe) (7) We provide a suitable introduction for meta-feature and embedding of AD component (from Reviewer n2uj). (8) We provide more details about the baselines of proposed meta-predictors to avoid confusion, such as the meaning of ‘GT’ or ‘SS’. (from Reviewer ufDM). (9) We revise Section 4.3, especially question 5, to clarify the pros and cons of the ML meta-predictors and DL meta-predictors.

- **Figures and Tables**: We have revised the captions of the tables to clarify the settings, as suggested by Reviewer ufDM. Additionally, we have enhanced the presentation of experimental results in Appx.  D by: (1) adding relative rankings in boxplots as additional indicators, (2) incorporating result tables on design choices across different domains, (3) including large benchmark evaluation result figures for synthetic datasets, (4) introducing results visualizations for unsupervised pre-trained models,  (5) presenting the performance visuals of ADGym's meta predictor on global anomalies, and (6) adding benchmark evaluation result figures under three noisy and corruption settings.

- **Additional Analysis**: We analyze: (1) the disadvantages of FTTransformer as a design choice, (2) the role of the Unsupervised pre-train model, (3) the performance of the design choice on specific anomaly datasets, (4) the performance of the design choice on domain-specific datasets, (5) the performance of the meta-predictor on specific anomaly datasets, (6) the performance of both the design choice and meta-predictor using the relative rank as a metric, and (7) the performance of the design choice under three noisy and corruption settings.

- **Future Directions**: We extend the future directions of ADGym in Section 5, e.g. analysis and decomposition of tree-based ML models (from Reviewer YA3B).


**Long-term plan**: We commit to maintaining and enriching ADGym in the long run, as many of our open-source AD works (e.g., PyOD [1], SUOD [2], TODS [3] and ADBench [4]). The next step is distributing ADGym via PyPI for easier access, as well as new design benchmarking. Also, we are considering developing ADGym according to our plan and the advice of reviewers in the future.

----

[1] Zhao, Yue, Zain Nasrullah, and Zheng Li. "PyOD: A Python Toolbox for Scalable Outlier Detection." JMLR, 2019.

[2] Zhao, Yue, et al. "SUOD: Accelerating large-scale unsupervised heterogeneous outlier detection." MLSys, 2021.

[3] Lai, Kwei-Herng, Daochen Zha, Junjie Xu, Yue Zhao, Guanchu Wang, and Xia Hu. "Revisiting time series outlier detection: Definitions and benchmarks." NeurIPS Benchmark and Datasets. 2021.

[4] Han, Songqiao, Hu, Xiyang, Huang, Hailiang, Jiang, Minqi, Zhao, Yue. "Adbench: Anomaly detection benchmark." Advances in Neural Information Processing Systems 35 (2022): 32142-32159.

---

> ### Author Response · Authors · 2023-08-26
> **Follow-up On Our Revision and Responses**
>
> We want to send this friendly reminder and are happy to address any further comments reviewers may have. Also, If our responses address your questions, we want to kindly ask for your reconsideration of the score.
>
> Thank you so much for devoting time to making ADGym a better work. We are looking forward to any further discussions :)

---

### Decision · Program_Chairs · 2023-09-22

**Decision:**

Accept (Poster)

**Comment:**

In this paper, the authors proposed  ADGym, a new benchmark and toolbox for anomaly detection (AD) algorithm evaluation and selection. The paper is well written,  the benchmark data is comprehensive.